# Quaternion Self-Attention with Shared Scores

**Shogo Yamauchi** [1] **Tohru Nitta** [2] **Hideaki Tamori** [1]

## Abstract

Quaternion neural networks are parameter-efficient and model multidimensional dependencies by representing four related features as a single entity. However, existing quaternion self-attention computes component-wise scores and applies independent softmax operations to each component, which increases the computational cost and allows attention distributions to diverge across components. We propose a shared-score quaternion self-attention mechanism that computes a single real-valued score using the quaternion inner product and applies a shared attention distribution across all components. This reduces score-computation multiplications by 75% and the number of softmax operations from four to one. We prove that, when queries and keys are produced by quaternion linear projections that induce component pre-mixing, the component-wise and shared scores lie in the same interaction subspace, indicating that independent component-wise attention primarily re-parameterizes the same interactions rather than expanding the feature interaction space. In speech enhancement, our method reduces inference time by up to 44.3% on a GPU and 58.1% on a CPU while maintaining quality, with consistent trends across vision and natural language processing. [2]

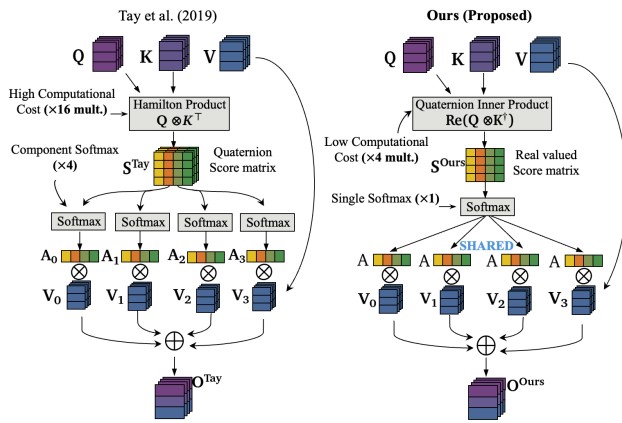

*Figure 1.* Comparison of quaternion self-attention architectures. **Left:** (Tay et al., 2019) computed the full Hamilton product using four independent softmax operations, which led to high computational cost and produced component-wise attention distributions. **Right (Ours):** Our method derives a single shared score via the quaternion inner product, reducing multiplications by 75% while preserving inter-component relationships.

## 1. Introduction

Quaternion neural networks (QNNs) leverage quaternions (Hamilton, 1847) to represent four related features

as a single entity, achieving high representational power with fewer parameters (Nitta, 1995; Gaudet & Maida, 2018; Isokawa et al., 2003; 2009; Yamauchi et al., 2025). Although QNNs have been applied to various tasks (Zhu et al., 2018; Hashim & Ogawa, 2022; Grant & Wang, 2025), designing quaternion self-attention mechanisms remains challenging in terms of computational efficiency and structural preservation.

Tay et al. (2019) computed a quaternion-valued score matrix using the Hamilton product and applied softmax independently to each component. This design required 16 real-valued multiplications per quaternion pair and four independent softmax operations. While such component-wise independent attention could theoretically enhance representational power by capturing diverse features across quaternion components, a fundamental question arises: Does this independence genuinely contribute to performance improvement?

Our analysis suggests otherwise. Specifically, we observe that the four components in the analysis by Tay et al. (2019) often attend to different positions, exhibiting only 3.83% inter-component top-1 (argmax) agreement. Although such

[1]The Asahi Shimbun Company, Tokyo, Japan [2]Tokyo Woman's Christian University, Tokyo, Japan. Correspondence to: Shogo Yamauchi <yamauchi-s1@asahi.com>, Tohru Nitta <tnitta@lab.twcu.ac.jp>, Hideaki Tamori <tamori-h@asahi.com>.

*Proceedings of the 43rd International Conference on Machine Learning*, Seoul, South Korea. PMLR 306, 2026. Copyright 2026 by the author(s).

[2]*https://github.com/asahi-research/Quaternion-Self-Attention-with-Shared-Scores*

diversity might intuitively be expected to enhance representational power, we found that it did not translate into measurable performance gains and might instead disrupt the intended quaternion coupling.

To address this redundancy and computational inefficiency, we propose the "shared-score quaternion self-attention mechanism" (Figure 1). Unlike the component-wise approach, our method computes a single attention score using the quaternion inner product, shared across all four components. This ensures consistent alignment while significantly reducing computational cost.

We validated our method primarily on single-channel speech enhancement. A central difficulty of this task is that the magnitude and phase of the short-time Fourier transform (STFT) spectrum must be estimated jointly; they are tightly coupled in the underlying complex spectrum, yet real-valued networks process them as independent channels and tend to lose this coupling. Quaternions are well suited to this regime, as the Hamilton product allows the magnitude and phase to be learned as a single coupled entity rather than as separate features. Real-time factor (RTF) is also a first-class concern in this domain, making efficient quaternion attention particularly valuable. To the best of our knowledge, this represents the first QNN application in this domain. The results demonstrate that a single shared score preserves performance, with consistent trends also observed on CIFAR-100 and SST-2.

Our contributions are as follows:

- **Identifying Structural Redundancy:** Under the pre-mixing induced by quaternion linear projections (Theorem C.1), the independent expressive power of the four components is structurally redundant and does not translate into measurable performance gains.
- **Efficient Shared-Score Mechanism:** We propose a quaternion self-attention mechanism based on the quaternion inner product. This design minimizes the number of score computation multiplications by 75% and softmax operations from four to one, by employing a single attention distribution shared across all components to preserve the intrinsic quaternion coupling.
- **Empirical Validation:** Our method reduces RTF by up to 44.3% on a GPU and 58.1% on a CPU over existing quaternion attention, with consistent trends across vision and natural language processing (NLP). We demonstrate that the single shared score closely approximates the attention patterns of computationally expensive baseline models while maintaining comparable performance.

## 2. Related Work

The quaternion attention scores approach, first formulated by Tay et al. (2019), incurred significant computational over-

head. Subsequent research (Chen et al., 2022; Yang et al., 2023; Zhou et al., 2024) deviated from this formulation by adopting quaternion convolution or rotational embedding. Although effective for local feature extraction or positional encoding, these approaches do not compute attention scores using the Hamilton product. Similarly, a recent study on pruning strategies for quaternion Transformers (Mukhopadhyay et al., 2024) employed quaternion algebra solely for parameter sharing in weight matrices, rather than for attention computation. Consequently, the algorithmic efficiency and theoretical properties of Hamilton-product-based attention have remained virtually unexplored since the study by Tay et al. (2019).

In the broader speech enhancement landscape, recent state-of-the-art models have leveraged state-space models (Chao et al., 2024) or specialized lightweight architectures (Yan et al., 2025). These advances target the SE backbone rather than the attention primitive itself, and are therefore orthogonal to the gap identified above. Our study addresses this gap by revisiting Hamilton-product-based attention and investigating whether component-wise independence is necessary.

## 3. Proposed Method

In this section, we formalize the proposed quaternion self-attention mechanism. Let $d_{\text{model}}$ and $d_h$ denote the quaternion feature dimensions of the model and each head, respectively, and let the number of heads be $H$. We employ $\otimes$ for the Hamilton product and $\cdot$ for the real-valued quaternion inner products.

### 3.1. Quaternion Algebra

A quaternion $q$ is defined by the real part $q_0$ and three imaginary parts $q_1, q_2, q_3$ as follows:

$$q = q_0 + q_1\mathsf{i} + q_2\mathsf{j} + q_3\mathsf{k}. \tag{1}$$

The set of all quaternions is denoted by $\mathbb{H}$. The imaginary units $\mathsf{i}$, $\mathsf{j}$, and $\mathsf{k}$ satisfy

$$\mathsf{i}^2 = \mathsf{j}^2 = \mathsf{k}^2 = \mathsf{ijk} = -1. \tag{2}$$

The conjugate and norm are denoted by

$$q^* = q_0 - q_1\mathsf{i} - q_2\mathsf{j} - q_3\mathsf{k}, \qquad \|q\|^2 = q \otimes q^*. \tag{3}$$

The Hamilton product $p \otimes q$ of two quaternions $p, q$ is expressed as

$$\begin{aligned} p \otimes q = &(p_0q_0 - p_1q_1 - p_2q_2 - p_3q_3) \\ &+ (p_0q_1 + p_1q_0 + p_2q_3 - p_3q_2)\mathsf{i} \\ &+ (p_0q_2 - p_1q_3 + p_2q_0 + p_3q_1)\mathsf{j} \\ &+ (p_0q_3 + p_1q_2 - p_2q_1 + p_3q_0)\mathsf{k}. \end{aligned} \tag{4}$$

## 3.2. Quaternion Inner Product

For quaternions $x = x_0 + x_1\mathsf{i} + x_2\mathsf{j} + x_3\mathsf{k}$ and $y = y_0 + y_1\mathsf{i} + y_2\mathsf{j} + y_3\mathsf{k} \in \mathbb{H}$, we define the **quaternion inner product (scalar product)** following Gürlebeck et al. (2007) as

$$x \cdot y := \frac{1}{2}\left(x \otimes y^* + y \otimes x^*\right)$$
$$= x_0 y_0 + x_1 y_1 + x_2 y_2 + x_3 y_3. \tag{5}$$

This inner product can be expressed as

$$x \cdot y = \mathrm{Re}(x \otimes y^*). \tag{6}$$

Specifically, consider the real part of the Hamilton product with the conjugate to be equivalent to the quaternion inner product. For quaternion vectors $\mathbf{q}, \mathbf{k} \in \mathbb{H}^d$, we define the inner product as the sum of the quaternion inner products over all components as follows:

$$\langle \mathbf{q}, \mathbf{k} \rangle := \sum_{\ell=1}^{d} q^{(\ell)} \cdot k^{(\ell)}$$
$$= \sum_{\ell=1}^{d} \left( q_0^{(\ell)} k_0^{(\ell)} + q_1^{(\ell)} k_1^{(\ell)} + q_2^{(\ell)} k_2^{(\ell)} + q_3^{(\ell)} k_3^{(\ell)} \right), \tag{7}$$

where $q^{(\ell)} = q_0^{(\ell)} + q_1^{(\ell)}\mathsf{i} + q_2^{(\ell)}\mathsf{j} + q_3^{(\ell)}\mathsf{k}$ and $k^{(\ell)} = k_0^{(\ell)} + k_1^{(\ell)}\mathsf{i} + k_2^{(\ell)}\mathsf{j} + k_3^{(\ell)}\mathsf{k}$.

This is equivalent to the Euclidean inner product in $\mathbb{R}^{4d}$ and satisfies (i) positive definiteness, (ii) symmetry, and (iii) bilinearity, establishing it as a mathematically valid similarity measure (see Appendix B.1).

## 3.3. Quaternion Self-Attention with Shared Scores

For a quaternion input $\mathbf{X} \in \mathbb{H}^{T \times d_{\mathrm{model}}}$ with sequence length $T$, we generate the queries, keys, and values via quaternion linear transformations (Appendix A.1) as follows:

$$\mathbf{Q} = \mathbf{X} \otimes \mathbf{W}_Q, \quad \mathbf{K} = \mathbf{X} \otimes \mathbf{W}_K, \quad \mathbf{V} = \mathbf{X} \otimes \mathbf{W}_V, \tag{8}$$

where $\mathbf{W}_Q, \mathbf{W}_K, \mathbf{W}_V \in \mathbb{H}^{d_{\mathrm{model}} \times d_h}$ are quaternion weight matrices corresponding to each head, and $d_h$ denotes the quaternion feature dimension per head. Let $\mathbf{Q}_i, \mathbf{K}_j, \mathbf{V}_j \in \mathbb{H}^{d_h}$ denote the quaternion vectors for the $i$-th and $j$-th tokens, respectively.

In our method, the score matrix $\mathbf{S} \in \mathbb{R}^{T \times T}$ is defined as

$$\mathbf{S} = \frac{1}{\sqrt{4d_h}} \mathrm{Re}(\mathbf{Q} \otimes \mathbf{K}^\dagger) \tag{9}$$

where $\mathbf{K}^\dagger := \mathbf{K}^{*\top}$ denotes the conjugate transpose, and $\mathrm{Re}(\cdot)$ denotes the real part applied element-wise to the

quaternion matrix. Here, the scaling factor $1/\sqrt{4d_h}$ is adopted because our formulation is mathematically equivalent to the standard scaled dot-product attention in the expanded real-valued space $\mathbb{R}^{4d_h}$, ensuring consistent variance.

Each element is expanded as follows:

$$\mathbf{S}_{ij} = \frac{1}{\sqrt{4d_h}} \langle \mathbf{Q}_i, \mathbf{K}_j \rangle = \frac{1}{\sqrt{4d_h}} \sum_{\ell=1}^{d_h} Q_i^{(\ell)} \cdot K_j^{(\ell)}. \tag{10}$$

The quaternion inner product is equivalent to the Euclidean inner product in $\mathbb{R}^{4d_h}$ (Eq. (7)), and Eq. (10) corresponds to the scaled dot-product of the real-valued expansions $\tilde{\mathbf{Q}}_i, \tilde{\mathbf{K}}_j \in \mathbb{R}^{4d_h}$, normalized by $\sqrt{4d_h}$, where $\tilde{\mathbf{Q}}_i = [Q_{i,0}^\top, Q_{i,1}^\top, Q_{i,2}^\top, Q_{i,3}^\top]^\top$ and $\tilde{\mathbf{K}}_j = [K_{j,0}^\top, K_{j,1}^\top, K_{j,2}^\top, K_{j,3}^\top]^\top$ denotes the concatenation of the four quaternion components.

This score aggregates interactions among all four quaternion components into a single scalar. Consequently, the alignment captures the aggregate similarity of the corresponding quaternion components using a single attention distribution shared across components. This simplification applies only to score computation; $\mathbf{Q}, \mathbf{K}, \mathbf{V}$ are generated by quaternion-parameterized projections with structured parameter sharing.

The attention weight matrix $\mathbf{A}^{\mathrm{ours}} \in \mathbb{R}^{T \times T}$ is computed by applying softmax row-wise as follows:

$$\mathbf{A}^{\mathrm{ours}} = \mathrm{softmax}(\mathbf{S}). \tag{11}$$

Crucially, $\mathbf{A}^{\mathrm{ours}}$ is a real-valued matrix shared across all quaternion components.

Expressing the value as

$$\mathbf{V} = \mathbf{V}_0 + \mathbf{V}_1\mathsf{i} + \mathbf{V}_2\mathsf{j} + \mathbf{V}_3\mathsf{k}, \tag{12}$$

the output $\mathbf{O}^{\mathrm{ours}} \in \mathbb{H}^{T \times d_h}$ is computed as

$$\mathbf{O}^{\mathrm{ours}} = \mathbf{A}^{\mathrm{ours}}\mathbf{V}_0 + \mathbf{A}^{\mathrm{ours}}\mathbf{V}_1\mathsf{i} + \mathbf{A}^{\mathrm{ours}}\mathbf{V}_2\mathsf{j} + \mathbf{A}^{\mathrm{ours}}\mathbf{V}_3\mathsf{k}. \tag{13}$$

Specifically, all four components share identical attention weights, preserving the quaternion structure, where the four components represent a single entity. Furthermore, softmax is computed only once on a $T \times T$ real-valued matrix. In multi-head attention, this computation is performed in parallel across all $H$ heads.

In contrast, Tay et al. (2019) computed a quaternion-valued score matrix $\mathbf{S}^{\mathrm{Tay}} \in \mathbb{H}^{T \times T}$ using the Hamilton product by applying softmax independently to each component as follows:

$$\mathbf{S}^{\mathrm{Tay}} = \frac{1}{\sqrt{d_h}} \mathbf{Q} \otimes \mathbf{K}^\top = \mathbf{S}_0 + \mathbf{S}_1\mathsf{i} + \mathbf{S}_2\mathsf{j} + \mathbf{S}_3\mathsf{k}, \tag{14}$$

| Method | Real multiplications | softmax |
|---|---|---|
| Tay et al. (2019) | 16 | 4 |
| Ours | 4 | 1 |

*Table 1.* Per-head computational cost for attention-score computation in quaternion self-attention. "Real multiplications" denotes the number of real-valued multiplications per quaternion pair used in the query–key interaction, and "softmax" denotes the number of independent softmax operations.

where $d_h$ denotes the quaternion feature dimension per head, i.e., $\mathbf{Q}_i, \mathbf{K}_j \in \mathbb{H}^{d_h}$, whose real-valued expansion has dimension $4d_h$. Our shared score is computed as a real-valued dot product in $\mathbb{R}^{4d_h}$ and therefore uses the standard scaling $1/\sqrt{4d_h}$, whereas Tay et al. (2019) scaled the quaternion-valued score by $1/\sqrt{d_h}$ and then applied softmax independently to each component.

$$\mathbf{A}_\alpha = \text{Componentsoftmax}(\mathbf{S}_\alpha^{\text{Tay}}), \quad \alpha \in \{0, 1, 2, 3\}, \tag{15}$$

where Componentsoftmax independently applies softmax to each quaternion component $(0, 1, 2, 3)$. The output is then computed as follows:

$$\mathbf{O}^{\text{Tay}} = \mathbf{A}_0\mathbf{V}_0 + \mathbf{A}_1\mathbf{V}_1\mathsf{i} + \mathbf{A}_2\mathbf{V}_2\mathsf{j} + \mathbf{A}_3\mathbf{V}_3\mathsf{k}. \tag{16}$$

This design has the following issues:

1. **Score computation:** Hamilton-product attention computes all four components, requiring 16 real-valued products per quaternion pair. In contrast, our method employs a single shared score $\text{Re}(\mathbf{Q} \otimes \mathbf{K}^\dagger)$, which is implemented as the sum of only four real matrix products (Eq. (9)).
2. **Softmax:** Prior methods apply softmax independently to four component-wise score matrices. In contrast, our approach simplifies this by applying a single softmax function to the shared real-valued score.
3. **Attention maps:** Component-wise attention learns distinct maps $\mathbf{A}_0, \mathbf{A}_1, \mathbf{A}_2, \mathbf{A}_3$, which allow the alignment to diverge. Our method mitigates this by sharing a single attention map across components, thereby preserving consistent quaternion coupling.

Table 1 summarizes the computational costs per head. Our method reduces both the number of real-valued multiplications for score computation and the number of softmax operations by 75%.

### 3.4. Query-Key Normalization

To control the scale of attention scores and preserve the structural integrity of quaternion features, we apply Quaternion RMSNorm (QRMSNorm) to the queries and keys. For a quaternion vector $\mathbf{q} = (q^{(1)}, \ldots, q^{(d)})^T \in \mathbb{H}^d$, we define the normalization for each component $q^{(\ell)} =$

$q_0^{(\ell)} + q_1^{(\ell)}\mathsf{i} + q_2^{(\ell)}\mathsf{j} + q_3^{(\ell)}\mathsf{k}$ as

$$\text{QRMSNorm}(q^{(\ell)}) = \frac{q^{(\ell)}}{\text{RMS}(q^{(\ell)})} \gamma^{(\ell)}, \tag{17}$$

where

$$\text{RMS}(q^{(\ell)}) = \sqrt{\frac{1}{4}\left((q_0^{(\ell)})^2 + (q_1^{(\ell)})^2 + (q_2^{(\ell)})^2 + (q_3^{(\ell)})^2\right) + \epsilon}, \tag{18}$$

and $\gamma^{(\ell)} \in \mathbb{R}$ is a learnable scalar gain. This normalization treats the four quaternion components as a single unit and stabilizes the attention scores by aligning the scales of the queries and keys.

## 4. Experiments

In this section, we evaluate the proposed shared-score quaternion attention mechanism by addressing three questions: (i) whether it matches the component-wise baseline in enhancement quality, (ii) whether it improves efficiency as measured by the RTF, and (iii) how quaternion models compare with real-valued baselines in parameter efficiency. Detailed training configurations, including short-time Fourier transform (STFT) parameters, optimizer settings, and RTF measurement protocol, are provided in Appendix D.3.

### 4.1. Datasets

To evaluate the effectiveness and scalability of the proposed method, we employed two standard benchmarks for speech enhancement. VoiceBank+DEMAND (Valentini-Botinhao et al., 2016) was employed to assess the effectiveness of quaternion-based speech enhancement (SE) through fair comparison with existing methods, whereas the DNS-Challenge 3 dataset (Reddy et al., 2021a) was used to evaluate the generalization performance and RTF under large-scale, diverse noise conditions.

**VoiceBank+DEMAND** (Valentini-Botinhao et al., 2016): We employed 11,572 utterances from 28 speakers in the VoiceBank corpus (Veaux et al., 2013) for training and 824 utterances from 2 unseen speakers for testing. Noise was selected from the DEMAND database (Thiemann et al., 2013) and mixed at SNRs of 0–15 dB during training, and at SNRs of 2.5–17.5 dB with unseen noise types during testing. All audio files were sampled at 16 kHz.

**DNS-Challenge 3 dataset** (Reddy et al., 2021a): This dataset was used to evaluate the generalization performance and RTF under large-scale, diverse noise conditions. The training data comprised 760 h of clean speech, 181 h of noise, and approximately 118,000 room impulse responses. The test set included multilingual utterances spanning various SNR and reverberation conditions. All audio was sampled at 16 kHz.

| Method | Attention Type | Params (M) | PESQ | CSIG | CBAK | COVL | STOI | SI-SDR |
|---|---|---|---|---|---|---|---|---|
| *Comparable Baselines (VoiceBank+DEMAND)* | | | | | | | | |
| Noisy | – | – | 1.97 | 3.35 | 2.44 | 2.63 | 0.91 | – |
| SEGAN (Pascual et al., 2017) | – | 97.47 | 2.16 | 3.48 | 2.94 | 2.80 | 0.92 | – |
| TSTNN (Wang et al., 2021) | – | 0.92 | 2.96 | 4.10 | 3.77 | 3.52 | 0.95 | – |
| MetricGAN+ (Fu et al., 2021) | – | – | 3.15 | 4.14 | 3.18 | 3.64 | 0.94 | – |
| SE-Conformer (Kim & Seo, 2021) | Standard | – | 3.13 | 4.45 | 3.55 | 3.82 | 0.95 | – |
| CMGAN (Cao et al., 2022) | Standard | 1.83 | 3.41 | 4.63 | 3.94 | 4.12 | 0.96 | – |
| DCCRN (Hu et al., 2020) | – | 3.07 | 2.59 | – | – | – | 0.94 | 18.74 |
| NSE-CATNet (Saleem et al., 2023) | Standard | 3.57 | 3.19 | 4.41 | 3.66 | 3.82 | 0.96 | – |
| Real-valued Conformer | Standard | 3.19 | 3.25 | 4.43 | 3.68 | 3.88 | 0.95 | 19.09 |
| *Quaternion* | | | | | | | | |
| QDenseNet | – (No Bottleneck) | 0.74 | 2.80 (2.75–2.85) | 3.84 | 3.41 | 3.32 | 0.94 | 18.42 (18.12–18.73) |
| QTN (Yang et al., 2023) | QConv-based attention | 0.63 | 2.76 (2.71–2.81) | 3.76 | 3.43 | 3.26 | 0.94 | 19.55 (19.29–19.82) |
| QTransformer (Tay et al., 2019) | Hamilton | 0.62 | 3.07 (3.02–3.12) | 4.26 | 3.61 | 3.68 | **0.95** | 19.62 (19.31–19.92) |
| QTransformer (Ours) | Shared Score | 0.62 | 3.07 (3.03–3.12) | 4.30 | 3.62 | 3.71 | **0.95** | **19.69** (19.45–19.94) |
| QConformer (Tay et al., 2019) | Hamilton | 0.80 | 3.11 (3.06–3.15) | 4.30 | 3.64 | 3.73 | **0.95** | 19.64 (19.39–19.90) |
| QConformer (Ours) | Shared Score | 0.80 | **3.18** (3.13–3.22) | **4.36** | **3.65** | **3.79** | **0.95** | 19.36 (19.08–19.66) |
| *Reference (Different Training Data / Foundation Models)* | | | | | | | | |
| DeepFilterNet3 (Schröter et al., 2023) | – | – | – | 3.17 | 4.34 | 3.61 | 3.77 | 0.94 | – |
| UNIVERSE++ (Scheibler et al., 2024) | – | 107.5 | 3.02 | – | – | – | 0.86 | 18.62 |

*Table 2.* Results of speech enhancement on VoiceBank+DEMAND. For PESQ and SI-SDR, we report the 95% bootstrap confidence intervals in parentheses (low–high). In the Attention Type column, "Standard" refers to real-valued attention. **Bold** denotes the best result among quaternion models. The bottom section lists recent models for reference, as they are trained on significantly larger datasets (e.g., DNS4) and employ foundation models.

## 4.2. Speech Enhancement Model

We applied the proposed shared-score quaternion self-attention mechanism to single-channel speech enhancement.

### 4.2.1. QUATERNION FEATURE REPRESENTATION

We applied the STFT to the input waveform $x(t)$ and denoted the complex spectrum at time frame $t$ and frequency bin $f$ as $X_{t,f} = \mathrm{Re}[X_{t,f}] + \mathrm{i}\,\mathrm{Im}[X_{t,f}] \in \mathbb{C}$. Following CMGAN (Cao et al., 2022), we adopted the amplitude, real part, and imaginary part as the three feature components. Following Parcollet et al. (2018); Muppidi & Radfar (2021), we constructed a pure imaginary quaternion

$$Q_{t,f} = 0 + |X_{t,f}|\,\mathsf{i} + \mathrm{Re}[X_{t,f}]\,\mathsf{j} + \mathrm{Im}[X_{t,f}]\,\mathsf{k}, \quad (19)$$

where $|X_{t,f}| = \sqrt{\mathrm{Re}(X_{t,f})^2 + \mathrm{Im}(X_{t,f})^2}$ is the amplitude spectrum.

### 4.2.2. ARCHITECTURE AND TRAINING

The model adopted an encoder-bottleneck-decoder architecture (see Figure 6 in Appendix D). The encoder and decoder consisted of a QDilated DenseNet with dilated convolutions. The bottleneck stacked $N$ layers of the quaternion Transformer or quaternion Conformer using our proposed method ($N = 2$ and heads $= 4$ in our experiments). Self-attention was applied along both the time and frequency dimensions to capture the global dependencies. For training, we employed a composite loss function combining multi-

resolution STFT, SI-SDR, complex L1, and PESQ losses. The details are provided in Appendix D.2.

## 4.3. Evaluation Metrics

We used the following metrics to evaluate speech enhancement performance. For VoiceBank+DEMAND, we report PESQ (Rix et al., 2001) (perceptual quality), CSIG, CBAK, COVL (Hu & Loizou, 2008) (predictors of subjective ratings), STOI (Taal et al., 2010) (intelligibility), and SI-SDR (Le Roux et al., 2019) (scale-invariant SDR). For DNS-Challenge, we report DNSMOS P.835 (Reddy et al., 2021b) (SIG, BAK, OVRL) and P.808. Computational efficiency was evaluated using RTF measured waveform-to-waveform on a single NVIDIA A100 80GB GPU and an Intel Xeon Gold 6342 CPU over 448 utterances (6.86k s) under matched implementation conditions for both methods. Furthermore, a detailed comparison of the computational complexity specific to the attention mechanism is provided in Appendix C.4.

## 4.4. Baselines and Ablations

Our primary comparison target is the quaternion attention of Tay et al. (2019). The real-valued Conformer is included as a reference point for measuring the parameter efficiency of quaternion representations, indicating whether the structural bias of quaternion models enables comparable quality with fewer parameters. For the VoiceBank+DEMAND dataset,

| Method | Attention Type | Params (M) | OVRL | SIG | BAK | P.808 | GPU RTF | CPU RTF |
|---|---|---|---|---|---|---|---|---|
| Noisy | – | – | 2.11 | 2.89 | 2.34 | 2.90 | – | – |
| Real-valued Conformer | Standard | 3.19 | 2.77 | 3.12 | 3.77 | 3.41 | 0.0071 | – |
| *Quaternion Models (Ours vs. Tay et al. (2019) Attention)* | | | | | | | | |
| QTransformer (Tay et al., 2019) | Hamilton | 0.62 | 2.61 (2.56–2.65) | 3.07 (3.02–3.11) | 3.44 (3.39–3.49) | 3.30 (3.26–3.33) | 0.0192 | 0.594 |
| QTransformer (Ours) | Shared Score | 0.62 | 2.67 (2.62–2.71) | 3.06 (3.01–3.10) | **3.61** (3.56–3.66) | **3.36** (3.32–3.40) | **0.0107** | **0.249** |
| QConformer (Tay et al., 2019) | Hamilton | 0.80 | 2.67 (2.62–2.71) | 3.08 (3.03–3.12) | 3.57 (3.52–3.62) | 3.33 (3.29–3.36) | 0.0202 | 0.610 |
| QConformer (Ours) | Shared Score | 0.80 | **2.69** (2.65–2.74) | **3.11** (3.06–3.15) | 3.57 (3.51–3.61) | 3.32 (3.29–3.36) | **0.0157** | **0.259** |

*Table 3.* Speech enhancement results on the DNS-Challenge 3 dataset. In the Attention Type column, "Standard" refers to real-valued attention. We report the 95% bootstrap confidence intervals over test utterances in parentheses for DNSMOS metrics (low–high). The best results among quaternion models with the same parameter count are shown in **bold**. Under matched implementation conditions, our method reduces RTF by up to 44.3% on GPU and 58.1% on CPU compared to the quaternion baseline (Tay et al., 2019), approaching real-valued efficiency. GPU RTF was measured on an NVIDIA A100 80GB; CPU RTF on an Intel Xeon Gold 6342.

we additionally evaluated QTN (Yang et al., 2023) to assess the effectiveness of query, key, and value computation versus quaternion convolutions, and a pure QDenseNet without the attention bottleneck to quantify the specific contribution of the self-attention layers.

### 4.5. Results

**VoiceBank+DEMAND.** Table 2 demonstrates that our proposed method maintains restoration quality comparable to the quaternion baseline (Tay et al., 2019) despite using a simplified attention mechanism. Specifically, our QConformer achieved a PESQ of 3.18, demonstrating consistent performance across CSIG and COVL and STOI close to the baseline. For SI-SDR, the shared-score approximation preserved signal fidelity within 0.3 dB. Compared to the real-valued Conformer, it retained 98% of the PESQ with only 25% of the parameters. Furthermore, it matches NSE-CATNet (Saleem et al., 2023) with $4.5\times$ fewer parameters and outperforms SE-Conformer and MetricGAN+. Although CMGAN (Cao et al., 2022) achieves higher PESQ, our method provides a favorable efficiency trade-off without requiring adversarial training. Finally, it substantially outperformed QTN and QDenseNet, validating the effectiveness of Hamilton-product-based global attention. For a qualitative comparison of spectral reconstruction, spectrogram visualizations are provided in Appendix C.6.

**DNS-Challenge 3 Dataset and Efficiency.** In DNS-Challenge 3 (Table 3), our method matched the baseline in quality while significantly reducing the runtime. Under matched implementation conditions, the GPU RTF decreased by 44.3% for our QTransformer ($1.79\times$ faster) and by 22.3% for our QConformer ($1.29\times$ faster); on the CPU, the reductions were 58.1% ($2.39\times$) and 57.5% ($2.36\times$), respectively. These gains can be attributed to the reduced score computation ($16\rightarrow4$ matmuls) and a single softmax, narrowing the efficiency gap to real-valued models.

**Summary.** Our method matches quaternion baselines in



*Figure 2.* Visualization of the first layer in the QTransformer bottleneck. The attention mechanism of Tay et al. (2019) attends to different positions for each component.

performance while substantially reducing the computational overhead and demonstrating superior parameter efficiency compared to real-valued models. These results indicate that, under quaternion linear projections, the additional expressiveness of independent component-wise softmax is unnecessary, positioning our shared-score mechanism as an efficient alternative.

## 5. Discussion: Why Does a Shared Score Suffice?

We propose a shared-score quaternion attention mechanism that employs a single scalar score (the quaternion inner product) to produce one attention distribution shared across components, achieving comparable or better performance while reducing the GPU RTF by up to 44.3% and the CPU RTF by up to 58.1%. In this section, we analyze the Voice-Bank+DEMAND models to explain why sharing a single

| Pair | Mean (%) | Std (%) |
|------|----------|---------|
| 0-1 | 5.64 | 13.69 |
| 0-2 | 3.45 | 9.91 |
| 0-3 | 5.62 | 17.22 |
| 1-2 | 2.14 | 6.33 |
| 1-3 | 4.16 | 12.36 |
| 2-3 | 1.97 | 6.49 |
| Overall | 3.83 | 6.58 |

*Table 4.* Component-wise agreement rates in the Hamilton product attention (Tay et al., 2019). Indices 0–3 denote quaternion components $(q_0, q_1, q_2, q_3)$. Agreement refers to the fraction of queries with matching argmax key positions between two components; the chance level is $\approx 1/T$ (0.63% by random sampling in Voice-Bank+DEMAND test set). The per-layer/head details are provided in Appendix C.5.

score is sufficient and to reveal the redundancy in Hamilton-product attention.

### 5.1. Analysis of Inter-Component Agreement

The key difference between the proposed method and the existing method (Tay et al., 2019) lies in the scope of the information used to compute the score matrix. In our method, the score matrix $\mathbf{S}$ is defined as the real part of the quaternion product and shares a single attention weight matrix across all components, yielding 100% inter-component consistency by design. In contrast, Tay et al. (2019) maintained independent attention weights for each quaternion component $(0, 1, 2, 3)$, ensuring that it theoretically possesses higher expressiveness than the former.

Our quantitative analysis revealed how this "expressiveness" was utilized. To measure the inter-component agreement introduced by Tay et al. (2019), we defined the agreement rate between the components as follows:

$$\text{Agreement}(m, n) = \frac{1}{T} \sum_{t=1}^{T} \mathbf{1}\Big[ \arg\max_s \mathbf{A}_m(t, s)$$
$$= \arg\max_s \mathbf{A}_n(t, s)\Big], \tag{20}$$

where $m, n \in \{0, 1, 2, 3\}$ denotes the quaternion components, $\mathbf{A}_m(t, s)$ is the attention weight from position $t$ to position $s$ for component $m$, and $T$ denotes the sequence length. In this analysis, four attention modules in the bottleneck (2 layers × {time, frequency}, 4 heads) of the trained QTransformer were utilized.

Table 4 shows the agreement rate for each component in the study by Tay et al. (2019), and Figure 2 shows the attention weight distribution of the first layer in time attention. As shown in Table 4, the average agreement rate across all pairs was only 3.83%, which is approximately six times

higher than the random baseline (0.63%) but significantly low in absolute terms. Thus, each component is focused on different positions.

Intuitively, component-wise independence might be expected to enhance representational power by capturing a diverse range of features. However, the comparable performance of our shared-score method suggests that this additional capacity for divergence is redundant for the task. As empirical performance alone is insufficient to justify redundancy, the next section presents theoretical and statistical analysis to explain this phenomenon.

### 5.2. Verification of Sufficient Expressiveness

Next, we discuss the mathematical rationale for why scores based on the quaternion inner product are sufficient for task performance. The score of the proposed method (Eq. (9)) is equivalent to the quaternion inner product (Eq. (6)), corresponding to the Euclidean inner product when the quaternion vectors are expanded to $\tilde{\mathbf{q}}, \tilde{\mathbf{k}} \in \mathbb{R}^{4d}$ (Eq. (7)). Specifically, the proposed method employs a hybrid design that leverages the quaternion structure (weight sharing) for feature extraction while utilizing Euclidean similarities for alignment. Our experimental results confirm that this design can produce outputs equivalent to those of existing methods.

1. **Distributional similarity:** The attention output distributions of our method and that of Tay et al. (2019) exhibit high similarity (KS statistic: 0.0128, Wasserstein distance: 0.028, quantile correlation: 0.995; Figure 3), confirming that projection onto the real part preserves the dynamic range required by subsequent layers.

2. **Approximation of attention outputs:** We next ask whether the two formulas, as mathematical functions, compute similar mappings. To isolate this question from any difference in learned representations, we extract $(Q, K, V)$ tensors from a trained model and pass them through both formulas. The resulting outputs are highly correlated: 0.78 when $(Q, K, V)$ is taken from our trained model, and 0.88 when taken from the trained model of Tay et al. (2019) (Figure 4(b),(c)).

3. **Independence of learned representations:** When trained independently, the two methods showed near-zero correlation (Pearson $r \approx 0.028$; Figure 4(a)), indicating convergence to distinct, yet functionally equivalent solutions.

These three results collectively demonstrate the following: Points 1 and 2 indicate that, despite the computational difference of the four-component independent softmax, its impact on the final output is limited. Point 3 shows that both methods achieve comparable performance while learning different representations, indicating that component-wise independent attention distributions are not essential for task performance. Therefore, the expressiveness of the exist-

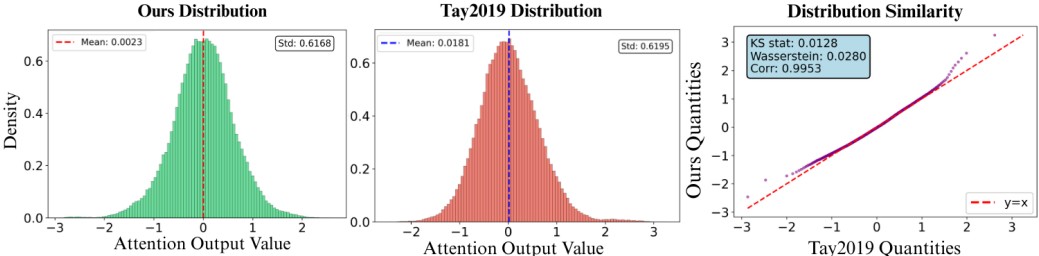

*Figure 3.* From left to right: Visualization of the attention output (flattened $\mathbf{O}^{\mathrm{ours}}$ values) distribution of the proposed method, the output distribution of Tay et al. (2019) (flattened $\mathbf{O}^{\mathrm{Tay}}$ values), and the quantile correlation between the two distributions. The results demonstrate a high degree of similarity, with a Kolmogorov–Smirnov (KS) statistic of 0.0128, a Wasserstein distance of 0.028, and an extremely high quantile correlation of 0.995.

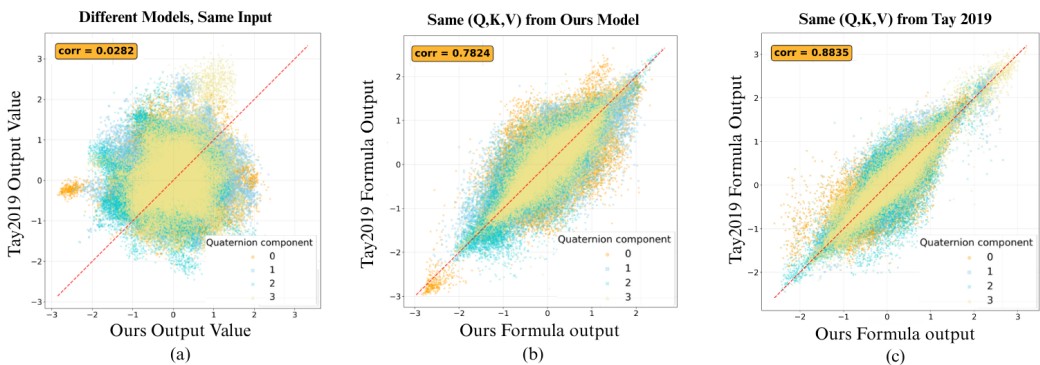

*Figure 4.* Correlation analysis of attention outputs. (a) Outputs from independently trained models on the same input show near-zero correlation (**corr** = 0.028). (b,c) Applying both formulas to identical $(Q, K, V)$ yields high correlation (**corr** = 0.78–0.88), indicating similar effective capacity despite different algebraic structures.

| Domain | Metric | Quality | | Efficiency Gain | Output Similarity | | |
| --- | --- | --- | --- | --- | --- | --- | --- |
| | | Tay et al. | Ours | | KS Stat. | Wasserstein | Quantile Corr. |
| Speech (DNS-3, QTransformer) | OVRL | 2.61 | 2.67 | GPU RTF $44.3\% \downarrow (1.79\times)$ | 0.0128 | 0.0280 | 0.9953 |
| Vision (CIFAR-100) | Acc. (%) | 71.09 | 70.94 | Training time $1.40\times$ | 0.0151 | 0.0203 | 0.9975 |
| NLP (SST-2) | Acc. (%) | 81.12 | 81.04 | Latency $26.8\% \downarrow (1.37\times)$ | 0.0407 | 0.0927 | 0.9961 |

*Table 5.* Cross-domain comparison of the results of the proposed shared-score attention against those of Tay et al. (2019). Quality is preserved within seed-level variance across all three domains while wall-clock cost is consistently reduced. Output-distribution similarity (KS statistic, Wasserstein distance, quantile correlation) is computed on the full attention outputs (after softmax and $\mathbf{AV}$ multiplication) between independently trained models. The quantile correlation exceeds 0.995 in every domain.

ing method, specifically the ability to "attend to different positions for each component," is largely superfluous.

### 5.3. Implications for Quaternion Self-Attention Design

The findings of this study are critical for the design of quaternion self-attention mechanisms. Various QNN architectures aim to preserve the Hamilton-product algebra throughout the computational graph. Our results indicate that, in architectures where queries and keys are produced by quaternion linear projections that already induce component pre-mixing, strictly maintaining component-wise hypercomplex

operations is unnecessary for attention-score computations.

The redundancy observed in this study is based on the strong inter-component coupling inherent in the quaternion linear layers. As detailed in Appendix A.1, a quaternion linear layer generates all four output components from a single shared weight matrix using the Hamilton product. Consequently, the resulting query $\mathbf{Q}$ and key $\mathbf{K}$ vectors are dense linear combinations of all the input components.

Our theoretical analysis (Appendix C.1) clarifies that prior coupling constrains the effective expressiveness of subsequent attention scores. Specifically, we prove that the four

component-wise score matrices in the existing method and the single shared score in our method are generated from the same interaction subspace $\mathcal{U}(\mathbf{W}_Q, \mathbf{W}_K)$ (Theorem C.1). Furthermore, because the components are already entangled by linear projections, separating the attention scores into four independent components does not expand the underlying feature interaction space; rather, it merely provides redundant parameterization within the same subspace.

This theoretical constraint also manifests empirically in the optimization dynamics. Our gradient-norm correlation analysis (Appendix C.3) reveals that inter-component gradient norms, which are nearly independent at random initialization, become substantially correlated after training (mean off-diagonal: 0.68). Thus, even when given independently learnable degrees of freedom, the optimization tends to couple the update magnitudes across components, consistent with the structural redundancy predicted by our analysis.

Therefore, we suggest that component interactions should be consolidated within the linear transformations. This motivates a "separation of concerns" principle: Hamilton products are employed in linear layers for structured feature coupling, while efficient quaternion inner products are used in attention to achieve consistent alignment. This design provides a practical balance between hypercomplex structural bias and computational efficiency.

### 5.4. Cross-Domain Validation

To verify that the redundancy claim is not specific to speech enhancement, we conducted additional experiments on image classification (CIFAR-100 (Krizhevsky, 2009)) and text classification (SST-2 (Socher et al., 2013)) under matched architecture capacity, batch size, and learning-rate scaling. The detailed configurations are presented in Appendix E.1 (CIFAR-100) and Appendix E.2 (SST-2).

Across all three domains, our shared-score method preserves task quality within seed-level variance while consistently reducing wall-clock cost (Table 5). On CIFAR-100, the accuracy is $70.94 \pm 0.24\%$ versus $71.09 \pm 0.34\%$ for the model of Tay et al. (2019), with a $1.40\times$ training-time speedup. On SST-2, the accuracy is $81.04 \pm 0.30\%$ versus $81.12 \pm 0.70\%$, while inference latency is reduced by $26.8\%$ ($1.37\times$ faster).

We further extended the output-distribution comparison from Figure 3 to all three domains. The right block of Table 5 reports the KS statistic, Wasserstein distance, and quantile correlation between the full attention outputs (including softmax and $\mathbf{AV}$ multiplication) of the two formulations. The quantile correlation exceeds 0.995 in every domain, indicating that the shapes of the two output distributions are nearly identical. The KS statistic and Wasserstein distance are slightly larger for NLP, but their absolute values remain small.

Taken together, these results indicate that the structural redundancy identified for component-wise attention on speech enhancement also holds across vision and NLP. The additional degrees of freedom of component-wise scoring do not translate into meaningfully different output distributions or task performance under the pre-mixing condition induced by quaternion linear projections (Theorem C.1). This supports the generality of the shared-score design across the three validated domains.

## 6. Conclusion

We investigated the necessity of component-wise independent attention in quaternion neural networks. Under the pre-mixing condition induced by quaternion linear projections, our analyses revealed that this independence is structurally redundant. Consequently, we proposed a shared-score mechanism utilizing the quaternion inner product, achieving up to 44.3% GPU and 58.1% CPU RTF reductions with comparable quality, and consistent trends across vision and NLP classification tasks.

Several limitations remain. The current evidence supports practical sufficiency only in the tested regimes; we have not validated larger models, longer-context tasks, or architectures without quaternion pre-mixing projections. More challenging robustness settings such as extreme reverberation, out-of-domain noise, and multi-speaker scenarios, as well as edge-device deployment and kernel-level runtime breakdowns, also remain tasks for the future. Beyond these, the theoretical scope of shared-score attention merits further study, particularly for tasks that require component-wise alignment (e.g., multimodal fusion or 3D rotation estimation). We will also explore hardware-aware implementations (e.g., FlashAttention (Dao et al., 2022)) and the early-stage optimization dynamics of component coupling.

## Impact Statement

This study proposes an efficient quaternion self-attention mechanism with applications in speech enhancement. Reducing the computational cost can facilitate deployment on resource-constrained devices, potentially improving the accessibility of speech enhancement technology. As is the case for various machine learning methods, potential risks include misuse (e.g., altering or fabricating audio) and biases inherited from the training data. Therefore, we do not anticipate any additional negative societal impact specific to our method beyond these general considerations.

## Acknowledgements

We thank the anonymous reviewers for their constructive comments, which substantially improved the paper.

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

# A. Quaternion Neural Network Layers

This section defines the quaternion neural network layers used in the proposed model. All layers possess a parameter-sharing structure based on the Hamilton product, which enforces strong coupling among the four components and achieves the same input-output dimensionality with approximately $1/4$ of the parameters compared to real-valued networks.

## A.1. Quaternion Linear Layer

The quaternion linear transformation layer operates on an input sequence $\mathbf{X} \in \mathbb{H}^{T \times d_{\text{in}}}$ and a quaternion weight matrix $\mathbf{W} \in \mathbb{H}^{d_{\text{in}} \times d_{\text{out}}}$ as follows:

$$\mathbf{Y} = \mathbf{X} \otimes \mathbf{W}. \tag{21}$$

Each element is expressed as

$$\mathbf{X} = \mathbf{X}_0 + \mathbf{X}_1 \mathsf{i} + \mathbf{X}_2 \mathsf{j} + \mathbf{X}_3 \mathsf{k}, \tag{22}$$

$$\mathbf{W} = \mathbf{W}_0 + \mathbf{W}_1 \mathsf{i} + \mathbf{W}_2 \mathsf{j} + \mathbf{W}_3 \mathsf{k}, \tag{23}$$

where $\mathbf{X}_0, \mathbf{X}_1, \mathbf{X}_2, \mathbf{X}_3 \in \mathbb{R}^{T \times d_{\text{in}}}$ and $\mathbf{W}_0, \mathbf{W}_1, \mathbf{W}_2, \mathbf{W}_3 \in \mathbb{R}^{d_{\text{in}} \times d_{\text{out}}}$.

Based on the Hamilton product, each output component is computed as follows:

$$\mathbf{Y}_0 = \mathbf{X}_0 \mathbf{W}_0 - \mathbf{X}_1 \mathbf{W}_1 - \mathbf{X}_2 \mathbf{W}_2 - \mathbf{X}_3 \mathbf{W}_3, \tag{24}$$

$$\mathbf{Y}_1 = \mathbf{X}_0 \mathbf{W}_1 + \mathbf{X}_1 \mathbf{W}_0 + \mathbf{X}_2 \mathbf{W}_3 - \mathbf{X}_3 \mathbf{W}_2, \tag{25}$$

$$\mathbf{Y}_2 = \mathbf{X}_0 \mathbf{W}_2 - \mathbf{X}_1 \mathbf{W}_3 + \mathbf{X}_2 \mathbf{W}_0 + \mathbf{X}_3 \mathbf{W}_1, \tag{26}$$

$$\mathbf{Y}_3 = \mathbf{X}_0 \mathbf{W}_3 + \mathbf{X}_1 \mathbf{W}_2 - \mathbf{X}_2 \mathbf{W}_1 + \mathbf{X}_3 \mathbf{W}_0. \tag{27}$$

Focusing on a single position (or feature dimension), let the input quaternion components be $\mathbf{x} = [x_0, x_1, x_2, x_3]^\top$ and the output be $\mathbf{y} = [y_0, y_1, y_2, y_3]^\top$. The quaternion linear transformation can be expressed in real matrix form as

$$\mathbf{y} = \begin{bmatrix} w_0 & -w_1 & -w_2 & -w_3 \\ w_1 & w_0 & w_3 & -w_2 \\ w_2 & -w_3 & w_0 & w_1 \\ w_3 & w_2 & -w_1 & w_0 \end{bmatrix} \begin{bmatrix} x_0 \\ x_1 \\ x_2 \\ x_3 \end{bmatrix}. \tag{28}$$

This matrix structure ensures that the four components are not transformed independently but are mutually coupled through shared weights $(w_0, w_1, w_2, w_3)$. While a real-valued linear layer requires $4d_{\text{in}} \times 4d_{\text{out}}$ parameters, the quaternion linear transformation layer requires only $4d_{\text{in}} \times d_{\text{out}}$ parameters, that is, one quarter of the parameters, a 75% reduction in the parameter count while simultaneously transforming all four components for the same input-output dimensionality.

## A.2. Quaternion Convolution Layer

The quaternion convolution layer extends the Hamilton product structure to convolutional kernels (Gaudet & Maida, 2018). Consider an input feature map $\mathbf{X} \in \mathbb{H}^{C_{\text{in}} \times H \times W}$ and a quaternion kernel $\mathbf{W} \in \mathbb{H}^{C_{\text{out}} \times C_{\text{in}} \times k \times k}$, each component of the output $\mathbf{Y} \in \mathbb{H}^{C_{\text{out}} \times H' \times W'}$ is computed as follows:

$$\mathbf{Y}_r = \mathbf{X}_r * \mathbf{W}_r - \mathbf{X}_i * \mathbf{W}_i - \mathbf{X}_j * \mathbf{W}_j - \mathbf{X}_k * \mathbf{W}_k, \tag{29}$$

$$\mathbf{Y}_i = \mathbf{X}_r * \mathbf{W}_i + \mathbf{X}_i * \mathbf{W}_r + \mathbf{X}_j * \mathbf{W}_k - \mathbf{X}_k * \mathbf{W}_j, \tag{30}$$

$$\mathbf{Y}_j = \mathbf{X}_r * \mathbf{W}_j - \mathbf{X}_i * \mathbf{W}_k + \mathbf{X}_j * \mathbf{W}_r + \mathbf{X}_k * \mathbf{W}_i, \tag{31}$$

$$\mathbf{Y}_k = \mathbf{X}_r * \mathbf{W}_k + \mathbf{X}_i * \mathbf{W}_j - \mathbf{X}_j * \mathbf{W}_i + \mathbf{X}_k * \mathbf{W}_r, \tag{32}$$

where the subscripts $r, i, j, k$ denote the real and imaginary parts, and $*$ denotes the standard real-valued convolution operation.

Similar to the quaternion linear layer, this structure enforces strong coupling among the four components compared to real-valued networks. Although a real-valued convolution layer with equivalent dimensions (i.e., $4C_{\text{in}}$ input and $4C_{\text{out}}$ output channels) requires $16C_{\text{in}}C_{\text{out}}k^2$ parameters, the quaternion convolution layer achieves the same dimensionality with $4C_{\text{in}}C_{\text{out}}k^2$ parameters. This yields a 75% reduction in the parameter count.

# B. Proof of Inner Product Properties

## B.1. Main Properties

**Proposition B.1.** *Based on the quaternion scalar product $x \cdot y$ defined in Eq. (5), we define the inner product for quaternion vectors $\mathbf{q}, \mathbf{k} \in \mathbb{H}^d$ as*

$$\langle \mathbf{q}, \mathbf{k} \rangle := \sum_{\ell=1}^{d} q^{(\ell)} \cdot k^{(\ell)} \in \mathbb{R}, \tag{33}$$

*where $q^{(\ell)}, k^{(\ell)} \in \mathbb{H}$ denote the $\ell$-th quaternion components. Then, $\langle \cdot, \cdot \rangle$ satisfies the following properties:*

1. ***Positive definiteness***: *$\langle \mathbf{q}, \mathbf{q} \rangle \geq 0$, with equality if and only if $\mathbf{q} = \mathbf{0}$.*

2. ***Symmetry***: *$\langle \mathbf{q}, \mathbf{k} \rangle = \langle \mathbf{k}, \mathbf{q} \rangle$.*

3. ***$\mathbb{R}$-bilinearity***: *For any $\alpha, \beta \in \mathbb{R}$,*

$$\begin{aligned}
\langle \alpha \mathbf{q}_1 + \beta \mathbf{q}_2, \mathbf{k} \rangle &= \alpha \langle \mathbf{q}_1, \mathbf{k} \rangle + \beta \langle \mathbf{q}_2, \mathbf{k} \rangle, \\
\langle \mathbf{q}, \alpha \mathbf{k}_1 + \beta \mathbf{k}_2 \rangle &= \alpha \langle \mathbf{q}, \mathbf{k}_1 \rangle + \beta \langle \mathbf{q}, \mathbf{k}_2 \rangle.
\end{aligned} \tag{34}$$

*Proof.* First, from Eq. (5), for any $x = (x_0, x_1, x_2, x_3), y = (y_0, y_1, y_2, y_3) \in \mathbb{H}$, we have

$$x \cdot y = x_0 y_0 + x_1 y_1 + x_2 y_2 + x_3 y_3. \tag{35}$$

Furthermore, from Eq. (6),

$$x \cdot y = \mathrm{Re}(x \otimes y^*). \tag{36}$$

Next, for $\mathbf{q} = (q^{(1)}, \dots, q^{(d)})$ with $q^{(\ell)} = (q_0^{(\ell)}, q_1^{(\ell)}, q_2^{(\ell)}, q_3^{(\ell)})$, we define the real vectorization $\tilde{\mathbf{q}} \in \mathbb{R}^{4d}$ as

$$\tilde{\mathbf{q}} := \left[ q_0^{(1)}, \dots, q_0^{(d)}, \ q_1^{(1)}, \dots, q_1^{(d)}, \ q_2^{(1)}, \dots, q_2^{(d)}, \ q_3^{(1)}, \dots, q_3^{(d)} \right]^\top, \tag{37}$$

and define $\tilde{\mathbf{k}}$. Then,

$$\begin{aligned}
\langle \mathbf{q}, \mathbf{k} \rangle &= \sum_{\ell=1}^{d} \left( q_0^{(\ell)} k_0^{(\ell)} + q_1^{(\ell)} k_1^{(\ell)} + q_2^{(\ell)} k_2^{(\ell)} + q_3^{(\ell)} k_3^{(\ell)} \right) \\
&= \tilde{\mathbf{q}}^\top \tilde{\mathbf{k}}.
\end{aligned} \tag{38}$$

Thus, $\langle \cdot, \cdot \rangle$ is equivalent to the standard inner product in $\mathbb{R}^{4d}$.

**(i) Positive definiteness**:

$$\langle \mathbf{q}, \mathbf{q} \rangle = \tilde{\mathbf{q}}^\top \tilde{\mathbf{q}} = \|\tilde{\mathbf{q}}\|_2^2 = \sum_{\ell=1}^{d} \left( (q_0^{(\ell)})^2 + (q_1^{(\ell)})^2 + (q_2^{(\ell)})^2 + (q_3^{(\ell)})^2 \right) \geq 0. \tag{39}$$

Equality holds if and only if $\tilde{\mathbf{q}} = \mathbf{0}$, that is, all components are zero, which implies $\mathbf{q} = \mathbf{0}$.

**(ii) Symmetry**:

$$\langle \mathbf{q}, \mathbf{k} \rangle = \tilde{\mathbf{q}}^\top \tilde{\mathbf{k}} = \tilde{\mathbf{k}}^\top \tilde{\mathbf{q}} = \langle \mathbf{k}, \mathbf{q} \rangle. \tag{40}$$

**(iii) $\mathbb{R}$-bilinearity**: This immediately follows from the bilinearity of the inner product in $\mathbb{R}^{4d}$.

**Remark.** In the multi-head attention described in Section 3.3, the query and key vectors for each head can be regarded as $\mathbf{q}, \mathbf{k} \in \mathbb{H}^{d_h}$. Therefore, this proposition directly applies to $d = d_h$. $\square$

## B.2. Necessity of Conjugate

Eq. (6) states that the quaternion scalar product is denoted by $\mathrm{Re}(q \otimes k^*)$. Here, we verify that $\mathrm{Re}(q \otimes k)$ without the conjugate is unsuitable as an inner product because:

Without the conjugate, the real part of the Hamilton product becomes

$$\mathrm{Re}(q \otimes k) = q_0 k_0 - q_1 k_1 - q_2 k_2 - q_3 k_3. \tag{41}$$

Consequently, the self-inner-product is

$$\mathrm{Re}(q \otimes q) = q_0^2 - q_1^2 - q_2^2 - q_3^2. \tag{42}$$

For example, for a pure imaginary quaternion $q = (0, 1, 0, 0)$, we have $\mathrm{Re}(q \otimes q) = -1 < 0$. Therefore, Eq. (42) does not satisfy the positive definiteness and is unsuitable as a similarity measure for the proposed method.

In contrast, when the conjugate is employed,

$$\mathrm{Re}(q \otimes q^*) = q_0^2 + q_1^2 + q_2^2 + q_3^2 = \|q\|^2 \geq 0, \tag{43}$$

This guarantees positive definiteness.

# C. Additional Analysis

## C.1. Theoretical Results

This section clarifies the relationship between the proposed method and the method developed by Tay et al. (2019) from two perspectives: (i) The generative structure of the score matrix (i.e., what interaction space is being explored), and (ii) The gradient flow from the loss to the scores (i.e., how learning signals are aggregated or separated).

### C.1.1. SCORE STRUCTURE UNDER QUATERNION LINEAR TRANSFORMATION

**Notation.** Let the components of the input $\mathbf{X} \in \mathbb{H}^{T \times d_{\mathrm{in}}}$ be denoted as $\mathbf{X} = (\mathbf{X}_0, \mathbf{X}_1, \mathbf{X}_2, \mathbf{X}_3)$, and similarly $\mathbf{Q} = (\mathbf{Q}_0, \mathbf{Q}_1, \mathbf{Q}_2, \mathbf{Q}_3)$. The quaternion linear transformations are given by $\mathbf{Q} = \mathbf{X} \otimes \mathbf{W}_Q$ and $\mathbf{K} = \mathbf{X} \otimes \mathbf{W}_K$ ($\mathbf{W}_Q, \mathbf{W}_K \in \mathbb{H}^{d_{\mathrm{in}} \times d_h}$). We also define the conjugate transpose as $\mathbf{K}^\dagger := (\mathbf{K}^*)^\top$.

**Score definitions.** Following our implementation, the score of the model of Tay et al. (2019) is defined as

$$\mathbf{S}^{\mathrm{Tay}} = \mathbf{Q} \otimes \mathbf{K}^\top, \tag{44}$$

where $\mathrm{softmax}$ is applied independently to each component $\mathbf{S}_\alpha^{\mathrm{Tay}}$ ($\alpha \in \{0, 1, 2, 3\}$). In contrast, the score of the proposed method is defined as

$$\mathbf{S} = \mathrm{Re}\big(\mathbf{Q} \otimes \mathbf{K}^\dagger\big), \tag{45}$$

with a single $\mathrm{softmax}$ function applied to Eq. (9). Note that even when considering a variant of Tay et al.'s formulation that employs conjugate $\mathbf{K}^*$, the signs in the equations change owing to the sign inversion of the imaginary parts; however, the decomposition structure in Theorem C.1 holds analogously. For simplicity, scaling factors are omitted in this section.

**Theorem C.1** (Score decomposition). *For an input $\mathbf{X} \in \mathbb{H}^{T \times d_{\mathrm{in}}}$, let $\mathbf{Q} = \mathbf{X} \otimes \mathbf{W}_Q$ and $\mathbf{K} = \mathbf{X} \otimes \mathbf{W}_K$. Then, each component $\mathbf{S}_\alpha^{\mathrm{Tay}}$ ($\alpha \in \{0, 1, 2, 3\}$) of the Hamilton product-based score $\mathbf{S}^{\mathrm{Tay}} = \mathbf{Q} \otimes \mathbf{K}^\top$ can be decomposed as follows:*

$$\mathbf{S}_\alpha^{\mathrm{Tay}} = \sum_{\beta, \gamma \in \{0,1,2,3\}} \mathbf{X}_\beta \, \mathbf{\Lambda}_\alpha^{\beta\gamma}(\mathbf{W}_Q, \mathbf{W}_K) \, \mathbf{X}_\gamma^\top, \tag{46}$$

*where $\mathbf{\Lambda}_\alpha^{\beta\gamma} \in \mathbb{R}^{d_{\mathrm{in}} \times d_{\mathrm{in}}}$ is the coefficient matrix determined by $\mathbf{W}_Q$ and $\mathbf{W}_K$.*

*Proof.* By the definition of the Hamilton product, each component of the quaternion linear transformation $\mathbf{Q} = \mathbf{X} \otimes \mathbf{W}_Q$ is given by

$$\mathbf{Q}_0 = \mathbf{X}_0 \mathbf{W}_{Q,0} - \mathbf{X}_1 \mathbf{W}_{Q,1} - \mathbf{X}_2 \mathbf{W}_{Q,2} - \mathbf{X}_3 \mathbf{W}_{Q,3}, \tag{47}$$

$$\mathbf{Q}_1 = \mathbf{X}_0 \mathbf{W}_{Q,1} + \mathbf{X}_1 \mathbf{W}_{Q,0} + \mathbf{X}_2 \mathbf{W}_{Q,3} - \mathbf{X}_3 \mathbf{W}_{Q,2}, \tag{48}$$

$$\mathbf{Q}_2 = \mathbf{X}_0 \mathbf{W}_{Q,2} - \mathbf{X}_1 \mathbf{W}_{Q,3} + \mathbf{X}_2 \mathbf{W}_{Q,0} + \mathbf{X}_3 \mathbf{W}_{Q,1}, \tag{49}$$

$$\mathbf{Q}_3 = \mathbf{X}_0 \mathbf{W}_{Q,3} + \mathbf{X}_1 \mathbf{W}_{Q,2} - \mathbf{X}_2 \mathbf{W}_{Q,1} + \mathbf{X}_3 \mathbf{W}_{Q,0}. \tag{50}$$

Thus, each component can be expressed in a unified form as

$$\mathbf{Q}_\mu = \sum_{\beta \in \{0,1,2,3\}} \mathbf{X}_\beta \, \boldsymbol{\Phi}_{\mu\beta}(\mathbf{W}_Q), \qquad \mathbf{K}_\nu = \sum_{\gamma \in \{0,1,2,3\}} \mathbf{X}_\gamma \, \boldsymbol{\Psi}_{\nu\gamma}(\mathbf{W}_K), \tag{51}$$

where $\boldsymbol{\Phi}_{\mu\beta}(\mathbf{W}_Q), \boldsymbol{\Psi}_{\nu\gamma}(\mathbf{W}_K) \in \mathbb{R}^{d_{\text{in}} \times d_h}$ are linear combinations of the components. Matrix $\boldsymbol{\Phi}(\mathbf{W}_Q)$ has the following structure:

$$\boldsymbol{\Phi}(\mathbf{W}_Q) = \begin{pmatrix} \mathbf{W}_{Q,0} & -\mathbf{W}_{Q,1} & -\mathbf{W}_{Q,2} & -\mathbf{W}_{Q,3} \\ \mathbf{W}_{Q,1} & \mathbf{W}_{Q,0} & \mathbf{W}_{Q,3} & -\mathbf{W}_{Q,2} \\ \mathbf{W}_{Q,2} & -\mathbf{W}_{Q,3} & \mathbf{W}_{Q,0} & \mathbf{W}_{Q,1} \\ \mathbf{W}_{Q,3} & \mathbf{W}_{Q,2} & -\mathbf{W}_{Q,1} & \mathbf{W}_{Q,0} \end{pmatrix}. \tag{52}$$

Matrix $\boldsymbol{\Psi}(\mathbf{W}_K)$ has the same structure.

Each component of the Hamilton product $\mathbf{S}^{\text{Tay}} = \mathbf{Q} \otimes \mathbf{K}^\top$ is given by

$$\mathbf{S}_0^{\text{Tay}} = \mathbf{Q}_0 \mathbf{K}_0^\top - \mathbf{Q}_1 \mathbf{K}_1^\top - \mathbf{Q}_2 \mathbf{K}_2^\top - \mathbf{Q}_3 \mathbf{K}_3^\top, \tag{53}$$

$$\mathbf{S}_1^{\text{Tay}} = \mathbf{Q}_0 \mathbf{K}_1^\top + \mathbf{Q}_1 \mathbf{K}_0^\top + \mathbf{Q}_2 \mathbf{K}_3^\top - \mathbf{Q}_3 \mathbf{K}_2^\top, \tag{54}$$

$$\mathbf{S}_2^{\text{Tay}} = \mathbf{Q}_0 \mathbf{K}_2^\top - \mathbf{Q}_1 \mathbf{K}_3^\top + \mathbf{Q}_2 \mathbf{K}_0^\top + \mathbf{Q}_3 \mathbf{K}_1^\top, \tag{55}$$

$$\mathbf{S}_3^{\text{Tay}} = \mathbf{Q}_0 \mathbf{K}_3^\top + \mathbf{Q}_1 \mathbf{K}_2^\top - \mathbf{Q}_2 \mathbf{K}_1^\top + \mathbf{Q}_3 \mathbf{K}_0^\top. \tag{56}$$

Substituting each $\mathbf{Q}_\mu \mathbf{K}_\nu^\top$ and rearranging, we obtain

$$\mathbf{Q}_\mu \mathbf{K}_\nu^\top = \sum_{\beta,\gamma} \mathbf{X}_\beta \, \boldsymbol{\Phi}_{\mu\beta}(\mathbf{W}_Q) \boldsymbol{\Psi}_{\nu\gamma}(\mathbf{W}_K)^\top \mathbf{X}_\gamma^\top. \tag{57}$$

Therefore, $\mathbf{S}_\alpha^{\text{Tay}}$ can be written as

$$\mathbf{S}_\alpha^{\text{Tay}} = \sum_{\beta,\gamma} \mathbf{X}_\beta \, \boldsymbol{\Lambda}_\alpha^{\beta\gamma} \, \mathbf{X}_\gamma^\top, \tag{58}$$

where

$$\boldsymbol{\Lambda}_\alpha^{\beta\gamma} := \sum_{(\mu,\nu) \in \mathcal{I}_\alpha} \sigma_{\alpha,\mu\nu} \, \boldsymbol{\Phi}_{\mu\beta}(\mathbf{W}_Q) \boldsymbol{\Psi}_{\nu\gamma}(\mathbf{W}_K)^\top. \tag{59}$$

Here, $\mathcal{I}_\alpha \subset \{0,1,2,3\}^2$ denotes the set of index pairs appearing in the expansion of $\mathbf{S}_\alpha^{\text{Tay}}$, and $\sigma_{\alpha,\mu\nu} \in \{+1,-1\}$ denotes the corresponding sign. $\qquad\square$

**Shared interaction subspace.** Theorem C.1 demonstrates that all score components can be expressed as sums of bilinear terms of the form $\mathbf{X}_\beta(\cdot)\mathbf{X}_\gamma^\top$. Each term combines two input components $(\mathbf{X}_\beta, \mathbf{X}_\gamma)$ through a coefficient matrix, where $\boldsymbol{\Phi}_{\mu\beta}(\mathbf{W}_Q)\boldsymbol{\Psi}_{\nu\gamma}(\mathbf{W}_K)^\top \in \mathbb{R}^{d_{\text{in}} \times d_{\text{in}}}$ serves as the fundamental building block of these coefficient matrices.

We define the **interaction subspace** as the subspace spanned by these coefficient matrices as follows:

$$\mathcal{U}(\mathbf{W}_Q, \mathbf{W}_K) := \text{span}\left\{ \boldsymbol{\Phi}_{\mu\beta}(\mathbf{W}_Q)\boldsymbol{\Psi}_{\nu\gamma}(\mathbf{W}_K)^\top \, \middle| \, \mu,\nu,\beta,\gamma \in \{0,1,2,3\} \right\} \subset \mathbb{R}^{d_{\text{in}} \times d_{\text{in}}}. \tag{60}$$

The discussion on the interaction subspace in this section addresses the score matrices prior to the application of the softmax function. The difference between the two methods manifests as a distinction in nonlinear mappings: whether softmax is applied once or four times to the same set of bilinear interactions.

**Corollary C.2** (Ours lies in the same interaction subspace). *The score of the proposed method,* $\mathbf{S} = \text{Re}(\mathbf{Q} \otimes \mathbf{K}^\dagger)$, *can also be expressed as*

$$\mathbf{S} = \sum_{\beta,\gamma \in \{0,1,2,3\}} \mathbf{X}_\beta \, \mathbf{M}^{\beta\gamma}(\mathbf{W}_Q, \mathbf{W}_K) \, \mathbf{X}_\gamma^\top. \tag{61}$$

*Furthermore, each* $\mathbf{M}^{\beta\gamma}$ *belongs to the same interaction subspace* $\mathcal{U}(\mathbf{W}_Q, \mathbf{W}_K)$ *defined in Eq. (60).*

*Proof.* From $\mathrm{Re}(q \otimes k^*) = q_0 k_0 + q_1 k_1 + q_2 k_2 + q_3 k_3$, we have

$$\mathbf{S} = \mathrm{Re}(\mathbf{Q} \otimes \mathbf{K}^\dagger) = \mathbf{Q}_0 \mathbf{K}_0^\top + \mathbf{Q}_1 \mathbf{K}_1^\top + \mathbf{Q}_2 \mathbf{K}_2^\top + \mathbf{Q}_3 \mathbf{K}_3^\top. \tag{62}$$

Expanding each $\mathbf{Q}_\alpha \mathbf{K}_\alpha^\top$ yields

$$\mathbf{Q}_\alpha \mathbf{K}_\alpha^\top = \sum_{\beta,\gamma} \mathbf{X}_\beta \, \mathbf{\Phi}_{\alpha\beta}(\mathbf{W}_Q) \mathbf{\Psi}_{\alpha\gamma}(\mathbf{W}_K)^\top \mathbf{X}_\gamma^\top. \tag{63}$$

Summing over all four components and rearranging, we obtain

$$\mathbf{S} = \sum_{\beta,\gamma} \mathbf{X}_\beta \, \mathbf{M}^{\beta\gamma} \, \mathbf{X}_\gamma^\top, \qquad \mathbf{M}^{\beta\gamma} := \sum_{\alpha \in \{0,1,2,3\}} \mathbf{\Phi}_{\alpha\beta}(\mathbf{W}_Q) \mathbf{\Psi}_{\alpha\gamma}(\mathbf{W}_K)^\top. \tag{64}$$

Because $\mathbf{M}^{\beta\gamma}$ is a linear combination of the generators in Eq. (60), it follows that $\mathbf{M}^{\beta\gamma} \in \mathcal{U}(\mathbf{W}_Q, \mathbf{W}_K)$. $\qquad\square$

**Remark.** We summarize the implications of Theorem C.1 and Corollary C.2.

**(1) Apparent complexity vs. actual simplicity:** The Hamilton product $\mathbf{Q} \otimes \mathbf{K}^\top$ is an operation that mixes components via bilinear interaction terms between the components. Intuitively, one might expect the four score components to capture different score contributions qualitatively. However, Theorem C.1 demonstrates that all components can be described as sums of terms with identical form $\mathbf{X}_\beta(\cdot)\mathbf{X}_\gamma^\top$.

**(2) Relationship with the proposed method:** Using Corollary C.2, the single score $\mathbf{S}$ of the proposed method is generated by coefficient matrices belonging to the same interaction subspace $\mathcal{U}(\mathbf{W}_Q, \mathbf{W}_K)$. Therefore, employing the four-component scores of the existing method can be understood as not expanding the interaction subspace itself, but rather as a design that increases the number of nonlinear mappings (component-wise softmax) applied to the same set of interactions.

**(3) Connection to empirical validation:** The experiments described in Section 5 confirmed that the agreement rate among the four components of the existing method was only 3.83%, indicating that each component learned substantially different attention patterns. Nevertheless, because the single score of the proposed method achieves comparable performance under this pre-mixing setting, the expressiveness introduced by component-wise softmax is redundant in the regime examined here.

### C.1.2. GRADIENT DYNAMICS

Next, we examine the gradient flow from the loss to score matrices.

**Theorem C.3** (Gradient aggregation vs. separation). *Assume that the loss function $\mathcal{L}$ is differentiable with respect to the output $\mathbf{O}$. In the proposed method, $\mathbf{A} = \mathrm{softmax}(\mathbf{S})$ is shared across all the components, with $\mathbf{O}_\alpha = \mathbf{A}\mathbf{V}_\alpha$ ($\alpha \in \{0,1,2,3\}$). Then,*

$$\frac{\partial \mathcal{L}}{\partial \mathbf{A}} = \sum_{\alpha \in \{0,1,2,3\}} \frac{\partial \mathcal{L}}{\partial \mathbf{O}_\alpha} \mathbf{V}_\alpha^\top, \tag{65}$$

$$\frac{\partial \mathcal{L}}{\partial \mathbf{S}} = J_{\mathrm{sm}}(\mathbf{S})^\top \left( \frac{\partial \mathcal{L}}{\partial \mathbf{A}} \right) = J_{\mathrm{sm}}(\mathbf{S})^\top \left( \sum_{\alpha \in \{0,1,2,3\}} \frac{\partial \mathcal{L}}{\partial \mathbf{O}_\alpha} \mathbf{V}_\alpha^\top \right), \tag{66}$$

*where $J_{\mathrm{sm}}(\mathbf{S})$ denotes the Jacobian (linear operator) of the row-wise softmax function. In contrast, the existing method computes $\mathbf{A}_\alpha = \mathrm{softmax}(\mathbf{S}_\alpha^{\mathrm{Tay}})$ independently for each component, with $\mathbf{O}_\alpha = \mathbf{A}_\alpha \mathbf{V}_\alpha$. Thus,*

$$\frac{\partial \mathcal{L}}{\partial \mathbf{S}_\alpha^{\mathrm{Tay}}} = J_{\mathrm{sm}}(\mathbf{S}_\alpha^{\mathrm{Tay}})^\top \left( \frac{\partial \mathcal{L}}{\partial \mathbf{O}_\alpha} \mathbf{V}_\alpha^\top \right), \qquad \alpha \in \{0,1,2,3\}, \tag{67}$$

*where each component score receives gradients only from its corresponding component output (i.e., the learning pathways are separated).*

*Proof.* Eq. (65) immediately follows the chain rule applied to $\mathbf{O}_\alpha = \mathbf{A}\mathbf{V}_\alpha$. Because softmax acts linearly on $\partial \mathcal{L}/\partial \mathbf{A}$, we have $\partial \mathcal{L}/\partial \mathbf{S} = J_{\mathrm{sm}}(\mathbf{S})^\top (\partial \mathcal{L}/\partial \mathbf{A})$, which yields Eq. (66). The derivation of Eq. (67) for the existing method is analogous. $\qquad\square$

**Proposition C.4** (Heuristic: gradient norm scale). *In the proposed method, the contributions from multiple components are aggregated into a single* $\mathbf{S}$ *using Eq.* (66). *When the statistics of* $\mathbf{V}_\alpha$ *and* $\partial\mathcal{L}/\partial\mathbf{O}_\alpha$ *are comparable across components,* $\|\partial\mathcal{L}/\partial\mathbf{S}\|_F$ *can be on the same order as* $\sum_\alpha\|\partial\mathcal{L}/\partial\mathbf{S}_\alpha^{\mathrm{Tay}}\|_F$. *We empirically verify this tendency in Table* 7.

### C.1.3. THEORETICAL IMPLICATIONS

By synthesizing the theoretical analysis and empirical validation presented above, we draw the following conclusions.

1. **Shared interaction subspace:** Both the four-component scores of the existing method and the single score of the proposed method are generated as linear combinations of bilinear interactions using coefficient matrices belonging to $\mathcal{U}(\mathbf{W}_Q, \mathbf{W}_K)$.

2. **Gradient aggregation vs. separation:** In the proposed method, the learning signals from all four components are aggregated into a single score, whereas in the existing method, independent gradient pathways emerge for each component.

3. **Implications for redundancy:** Score generation is reduced to reweighting within the same interaction set; however, as softmax is nonlinear, equivalence at the attention output level cannot be established from this analysis alone. We therefore empirically verify post-softmax similarity in Figure 3 and Figure 4(b,c), which together with the shared interaction subspace provide the basis for the limited effective contribution of component-wise softmax.

### C.2. Gradient Flow Analysis

This section provides an empirical verification of Theorem C.3. We compared the gradient flow in the proposed method with that in the method proposed by Tay et al. (2019) and confirmed that the theoretical analysis was reflected in the computational graph.

**Experimental setup.** We randomly initialized the quaternion self-attention layer and analyzed the gradient flow. The experimental conditions are listed in Table 6.

| Symbol | Description | Value |
|---|---|---|
| $B$ | Batch size | 32 |
| $T$ | Sequence length | 128 |
| $D_{model}$ | Embedding dimension | 64 |
| $N_{\mathrm{trial}}$ | Number of independent trials | 100 |

*Table 6.* Experimental setup for gradient analysis.

where $\mathbf{S} \in \mathbb{R}^{T \times T}$ denotes the attention-score matrix of the proposed method (Eq. (9)), and $\mathbf{S}_0^{\mathrm{Tay}}, \mathbf{S}_1^{\mathrm{Tay}}, \mathbf{S}_2^{\mathrm{Tay}}, \mathbf{S}_3^{\mathrm{Tay}} \in \mathbb{R}^{T \times T}$ denotes the attention-score matrices for each quaternion component in Tay et al. (2019). We measured the gradient norm $\|\partial\mathcal{L}/\partial\mathbf{S}\|_F$ (Frobenius norm) from loss function $\mathcal{L}$ to each score matrix.

**Verification of Theorem C.3.** Table 7 presents a comparison of the gradient norms.

| Method | Target | Gradient Norm |
|---|---|---|
| Ours | Single $\mathbf{S}$ | $(1.72 \pm 0.02) \times 10^{-3}$ |
| Tay et al. (2019) | $\mathbf{S}_0^{\mathrm{Tay}}$ | $(4.86 \pm 0.07) \times 10^{-4}$ |
| | $\mathbf{S}_1^{\mathrm{Tay}}$ | $(4.87 \pm 0.06) \times 10^{-4}$ |
| | $\mathbf{S}_2^{\mathrm{Tay}}$ | $(4.87 \pm 0.06) \times 10^{-4}$ |
| | $\mathbf{S}_3^{\mathrm{Tay}}$ | $(4.88 \pm 0.06) \times 10^{-4}$ |

*Table 7.* Gradient norms for attention-score matrices.

The gradient norm of the proposed method ($1.72 \times 10^{-3}$) is of the same order as the sum of the gradient norms of the four components in the model proposed by Tay et al. (2019) (($4.86 + 4.87 + 4.87 + 4.88) \times 10^{-4} = 1.95 \times 10^{-3}$), which is consistent with Theorem C.3, where the gradients are aggregated into a single score matrix.

**Remark.** Although the mathematical content of Theorem C.3 is an application of the chain rule, its significance lies in its design implications for quaternion neural networks. In gradient aggregation of the proposed method (Eq. (66)), all four components send learning signals to a single score $\mathbf{S}$; consequently, conflicting demands across components are averaged, and *consistent alignment* is enforced mathematically. However, in the gradient separation in the study by Tay et al. (2019) (Eq. (67)), only $\mathbf{O}_0$ contributes to the gradient of $\mathbf{S}_0$, only $\mathbf{O}_1$ contributes to the gradient of $\mathbf{S}_1$, etc. Consequently, each component is optimized independently without considering the outputs of the other components, which can lead to learning attention distributions that focus on different positions across components (agreement rate of 3.83% in Table 4). This property partially explains why performance is maintained even when the four independent component scores are simplified to a single score (Table 2).

### C.3. Gradient Correlation Analysis

This section analyzes the correlation between the gradients and component-wise score matrices in the model of Tay et al. (2019). We conducted experiments on randomly initialized and trained models to observe changes through learning.

**Experimental setup.** The same settings as those listed in Table 6 were used. The input data were randomly generated for both experiments, and the gradient norm correlations were measured by varying the model weights.

**Results.** Table 8 presents the gradient norm correlation matrices for the randomly initialized model (a) and the trained model (b).

| | 0 | 1 | 2 | 3 | | 0 | 1 | 2 | 3 |
|---|---|---|---|---|---|---|---|---|---|
| 0 | 1.000 | $< 10^{-3}$ | $< 10^{-3}$ | $< 10^{-3}$ | 0 | 1.000 | 0.826 | 0.317 | 0.879 |
| 1 | $< 10^{-3}$ | 1.000 | $< 10^{-3}$ | $< 10^{-3}$ | 1 | 0.826 | 1.000 | 0.792 | 0.833 |
| 2 | $< 10^{-3}$ | $< 10^{-3}$ | 1.000 | $< 10^{-3}$ | 2 | 0.317 | 0.792 | 1.000 | 0.404 |
| 3 | $< 10^{-3}$ | $< 10^{-3}$ | $< 10^{-3}$ | 1.000 | 3 | 0.879 | 0.833 | 0.404 | 1.000 |
| | (a) Random initialization | | | | | (b) After training | | | |

*Table 8.* Gradient norm correlation matrices for the model of Tay et al. (2019). Indices $0, 1, 2, 3$ denote the quaternion components. (a) At random initialization, the gradient norms are nearly independent. (b) After training on VoiceBank+DEMAND, a strong correlation emerges (mean off-diagonal: 0.68). Both experiments use **random input data**, *largely* isolating the effect of learned weights.

For random initialization (Table 8(a)), all off-diagonal elements were $< 10^{-3}$, confirming that the component-wise gradients were nearly independent of each other. This is consistent with the analysis in Theorem C.3, which indicates that the design of Tay et al. (2019) separates the gradient pathways.

In contrast, for the trained model (Table 8(b)), **the inter-component gradient norm correlation increased to a mean of 0.68, despite the random input**. This indicates that through training, the weights $\mathbf{W}_Q, \mathbf{W}_K$ acquired a correlation structure among the components.

**Design implications.** These provide important insights regarding the design of Tay et al. (2019):

1. **Gradient separation by design:** The low correlation ($< 10^{-3}$) at random initialization confirms that, by design, the gradient pathways to each $\mathbf{S}_\alpha^{\text{Tay}}$ are separated, as shown in Theorem C.3.

2. **Acquisition of the correlation structure through learning:** However, after training, this correlation increased to 0.68. Because the input is random, this correlation does not depend on the input distribution but is a structure embedded in the learned weights $\mathbf{W}_Q, \mathbf{W}_K$. Although the four components have degrees of freedom that allow independent learning, they converge to representations that receive updates of similar magnitude.

3. **Evidence consistent with redundancy:** A design intended to express component-independent attention distributions can become structurally redundant. Therefore, the expressive power available for independent learning may not be utilized independently.

4. **Consistency with agreement rate:** The gradient norm correlation after training (0.68) and the inter-component agreement rate (3.83%, Table 4) appear contradictory at first, but this can be interpreted as follows:

- High gradient norm correlation: The update magnitudes for the four components co-vary and become strongly coupled.
- Low agreement rate (Tables 4 and 10): However, owing to the independent softmax for each component, the actual maximum attention positions (argmax) can be distributed across different tokens.

In component-wise attention, even when component updates are coupled in magnitude, the independent score matrices $\mathbf{S}_\alpha^{\text{Tay}}$ and component-wise softmax can preserve differences in score space as differences in attention distributions.

This property partially explains why the proposed method can achieve a comparable or superior performance while reducing the number of scores to a single score.

## C.4. MACs and Runtime Analysis

| $T$ | Ours | | Tay et al. (2019) | | Speedup |
|---|---|---|---|---|---|
| | MACs | Time [ms] | MACs | Time [ms] | |
| 512 | 134M | 1.210 | 336M | 1.536 | 1.27× |
| 1024 | 537M | 1.842 | 1.34G | 3.038 | 1.65× |
| 2048 | 2.15G | 5.251 | 5.37G | 9.780 | 1.86× |
| 4096 | 8.59G | 15.500 | 21.5G | 32.231 | 2.08× |

*Table 9.* Computational cost and wall-clock latency of a single quaternion self-attention layer ($D_{model}$=64, $H$=8). Latency is measured under matched implementation conditions on an NVIDIA A100 80GB PCIe with batch size 1, using `torch.cuda.Event` after 50 warmup iterations; values are median over 200 runs.

We compared the theoretical computational cost (MACs) and measured the inference time for a single quaternion self-attention layer. The evaluation was conducted on an NVIDIA A100 80GB PCIe with a batch size of 1, measuring only the forward pass. Because GPU execution is asynchronous, timing measurements were performed using `torch.cuda.Event` with synchronization after 50 warmup iterations, and we report the median over 200 runs (standard deviations are shown in parentheses). We fixed the embedding dimension at $D_{model}$=64 and the number of heads at $H$=8 and evaluated the scaling behavior across sequence lengths $T \in \{512, 1024, 2048, 4096\}$.

As shown in Table 9, the proposed method reduces the theoretical MACs by an average of 60% compared with the method of Tay et al. (2019), achieving speedups of $1.27\times$ to $2.08\times$ in wall-clock time. This improvement stems from the proposed shared-score mechanism. In contrast to the standard quaternion attention, which computes four component-wise score matrices (equivalent to 16 real-valued matmuls for the query–key interaction), our approach computes a single shared attention map. This design reduces the number of softmax normalizations from four (one per quaternion component) to a single shared softmax function. In terms of real-valued matrix multiplications for score computation, it reduces the cost from 16 matmuls (full Hamilton product) to 4 matmuls (quaternion inner product), that is, a theoretical reduction of 75%. Note that the measured wall-clock time does not scale strictly proportionally with the theoretical MACs owing to factors such as the GPU kernel selection and memory bandwidth constraints.

## C.5. Detailed Agreement Analysis

Table 10 presents the agreement rates for each layer-head combination. The frequency-direction layers (mean 6.58%) exhibit higher agreement rates than the time-direction layers (mean 1.08%), but both remain at low levels. The expected value under random selection is $1/T$, which depends on the sequence length $T$. To assess robustness beyond Top-1, we also measured Top-5 agreement (Table 11). The observed rate of 7.29% remains low (chance level: 3.16%), suggesting that the component-wise attention often selects different argmax positions even when considering multiple high-attention candidates.

## C.6. Qualitative Comparison on Spectral Reconstruction

To further validate the effectiveness of the proposed method, we conducted a qualitative analysis of the spectrograms on the VoiceBank+DEMAND test set. Figure 5 presents a visual comparison between the proposed Shared-Score Quaternion Con-

| Frequency layers | | | Time layers | | |
|---|---|---|---|---|---|
| Layer-Head | Mean (%) | Std (%) | Layer-Head | Mean (%) | Std (%) |
| freq_layers.0.head0 | 10.49 | 10.69 | time_layers.0.head0 | 0.35 | 0.55 |
| freq_layers.0.head1 | 2.55 | 1.97 | time_layers.0.head1 | 0.32 | 0.91 |
| freq_layers.0.head2 | 1.78 | 2.64 | time_layers.0.head2 | 0.33 | 0.41 |
| freq_layers.0.head3 | 8.49 | 5.37 | time_layers.0.head3 | 0.24 | 0.44 |
| freq_layers.1.head0 | 16.23 | 11.91 | time_layers.1.head0 | 1.77 | 2.42 |
| freq_layers.1.head1 | 2.52 | 2.97 | time_layers.1.head1 | 3.26 | 3.45 |
| freq_layers.1.head2 | 6.83 | 6.63 | time_layers.1.head2 | 1.49 | 2.60 |
| freq_layers.1.head3 | 3.78 | 3.54 | time_layers.1.head3 | 0.85 | 1.38 |
| Overall | | | 3.83 (%) | | |

*Table 10.* Per layer-head agreement rates in the Hamilton product attention (Tay et al., 2019).

| **Metric** | **Observed** | **Chance** ($K/T$) | **Ratio** |
|---|---|---|---|
| Top-1 Agreement | 3.83% | 0.63% | 6.07$\times$ |
| Top-5 Agreement | 7.29% | 3.16% | 2.31$\times$ |

*Table 11.* Inter-component agreement in Tay et al. (2019).

former (Ours) and three baselines: the QTN (Yang et al., 2023) and Real-valued Conformer, and Quaternion Conformer (Tay et al., 2019).

**Comparison with Real-Valued Baseline.** This example presents the advantage of quaternion representations in suppressing stationary low-frequency noise. Specifically, at approximately 64 Hz, the real-valued Conformer leaves visible residual energy that resembles a noisy input (Figure 5, bottom-center). In contrast, our shared-score quaternion Conformer produces a cleaner background that is closer to the clean reference (top-left). This observation is consistent with the intuition that inter-component coupling in quaternion models can help capture dependencies among related features.

**Comparison with QTN.** Although QTN (Yang et al., 2023) suppresses noise, it appears overly aggressive in this case. The low-frequency region is noticeably darker than the clean reference, suggesting over-suppression, where speech components may be eliminated together with noise. This may be related to QTN's reliance on local convolution operations, which can be less effective at leveraging long-range temporal context than attention-based models. These observations align with the lower objective scores (PESQ 2.76) reported in Table 2.

**Comparison with Component-wise Attention.** Finally, our output is highly similar to that of the computationally expensive model of Tay et al. (2019). Both quaternion attention methods maintain a good balance between noise suppression and signal preservation, closely matching the spectral characteristics of the clean target. This qualitative evidence supports our quantitative findings that a single shared score can be sufficient in this setting while reducing the score-computation cost by 75%.

## D. Experimental Details

### D.1. Model Architecture

The model adopts an encoder–bottleneck–decoder architecture (Figure 6(a)).

**Encoder.** The encoder consists of a QDilated DenseNet (Figure 6(c)). It assumes quaternion features $\mathbf{X} \in \mathbb{H}^{F \times T}$ as input and densely stacks QConv–QBatchNorm–PReLU–dropout layers with dilation rates $d \in \{1, 2, 4\}$. Similar to the WaveNet (van den Oord et al., 2016) architecture, we introduced a sigmoid-based gating mechanism to achieve a wide receptive field and selective feature propagation. The output was down-sampled along the frequency axis to yield $\mathbb{H}^{C \times (F/2) \times T}$.

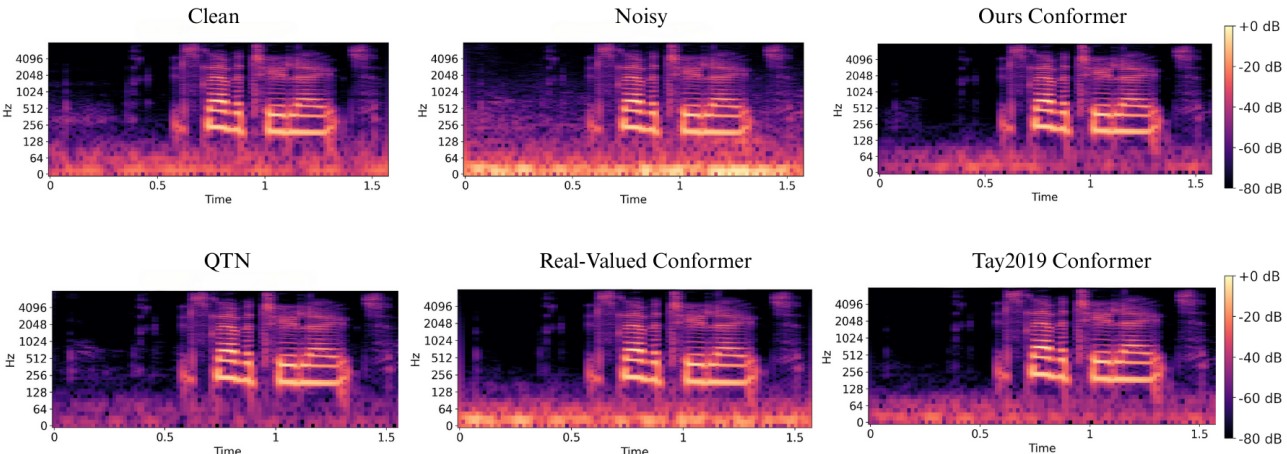

*Figure 5.* Log-magnitude spectrogram comparison on VoiceBank+DEMAND (a representative test utterance; all panels share the same color scale). The real-valued Conformer leaves residual low-frequency noise, whereas QTN (Yang et al., 2023) tends to over-suppress low-frequency components. Both quaternion attention methods (ours and that of Tay et al. (2019)) are visually close to the clean reference.

**Bottleneck.** The bottleneck stacks $N$ layers of quaternion Transformer or quaternion Conformer blocks using the proposed shared-score quaternion self-attention to capture global dependencies in both the time and frequency directions (Figure 6(b)). The quaternion Transformer is based on a Transformer (Vaswani et al., 2017) and consists of QLayerNorm–QMHA–QFFN. The quaternion Conformer follows the Conformer (Gulati et al., 2020) architecture, augmenting it with a quaternion convolution module (QConv Module) to complement local patterns. To apply self-attention in both the time and frequency directions, the tensor was reshaped to $[B \times F, T, C]$ and $[B \times T, F, C]$, and the same block was applied to each.

**Decoder.** The decoder consists of a QDilated DenseNet, but with dilation rates $d \in \{8, 4, 2\}$, starting from a large receptive field and gradually recovering from finer structures. Upsampling is performed along the frequency axis to restore the dimensions from $F/2$ to the original $F$. Finally, a two-channel real-valued convolution layer outputs the magnitude mask $M_{\mathrm{mag}}$ and the phase correction $M_{\mathrm{phase}}$, and the enhanced waveform is reconstructed by applying the inverse STFT to the enhanced spectrum obtained as

$$|\hat{S}_{t,f}| = M_{\mathrm{mag},t,f} \cdot |X_{t,f}|, \tag{68}$$

$$\angle\hat{S}_{t,f} = \angle X_{t,f} + M_{\mathrm{phase},t,f}. \tag{69}$$

### D.2. Training Loss

We employed a composite loss function that considered the time, frequency, and perceptual quality aspects. The overall loss $\mathcal{L}$ is defined as the following weighted sum:

$$\mathcal{L} = 3.5\,\mathcal{L}_{\mathrm{MSTFT}} + 2.0\,\mathcal{L}_{\mathrm{SI\text{-}SDR}} + \mathcal{L}_{\mathrm{RMS}} + 2.5\,\mathcal{L}_{\mathrm{CL1}} + \mathcal{L}_{\mathrm{PESQ}}. \tag{70}$$

The loss weights were determined via a small validation sweep to maximize the PESQ of the validation set. Here, $x, \hat{x} \in \mathbb{R}^N$ denote the target and estimated time-domain waveforms, respectively, and $\mathbf{S}, \hat{S} \in \mathbb{C}^{F \times T}$ denote the target and estimated complex spectrograms, respectively.

**Multi-Resolution STFT Loss (Yamamoto et al., 2020).** This loss is defined as the sum of the spectral convergence loss and magnitude loss across different FFT sizes $\{512, 1024, 2048\}$, which helps improve the reconstruction accuracy at multiple time-frequency resolutions.

**SI-SDR Loss (Li et al., 2020).** This loss guides the training to maximize the scale-invariant signal-to-distortion ratio (SI-SDR), thereby improving waveform fidelity in the time domain.

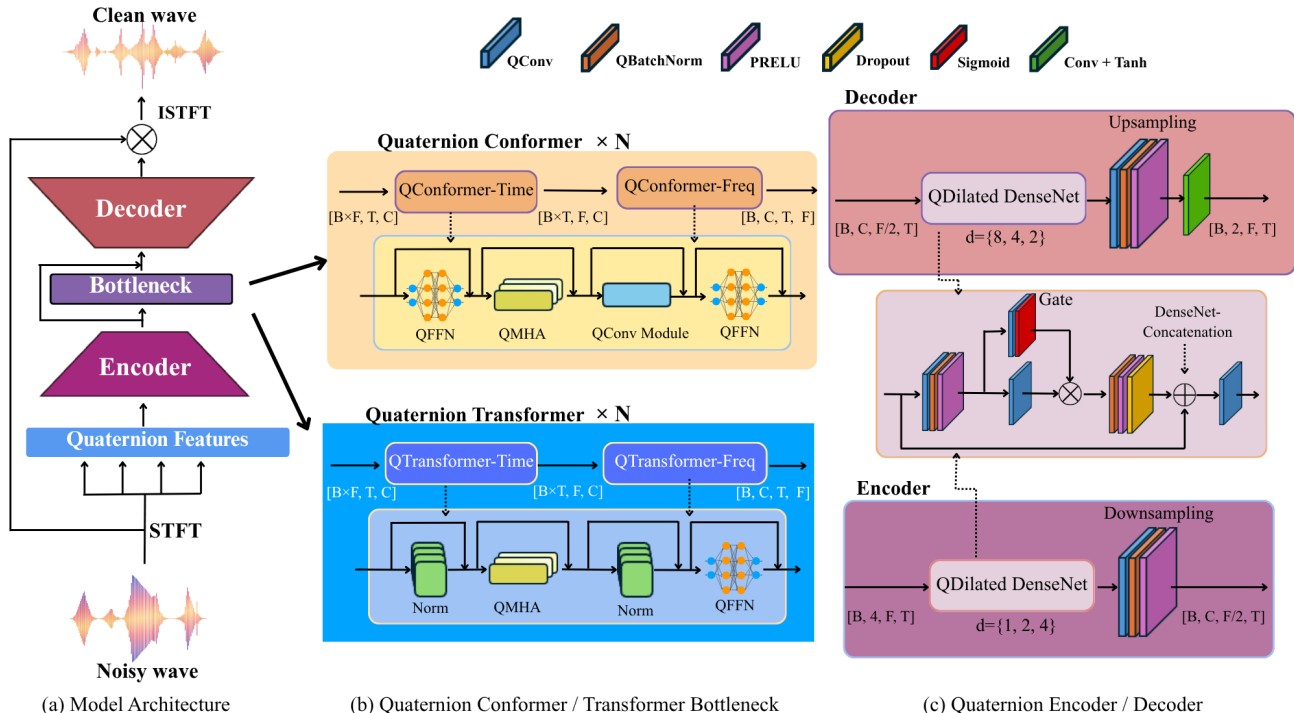

(a) Model Architecture  (b) Quaternion Conformer / Transformer Bottleneck  (c) Quaternion Encoder / Decoder

*Figure 6.* Overview of the proposed quaternion speech enhancement model. (a) Overall encoder–bottleneck–decoder architecture. (b) Quaternion Transformer/Conformer block with the proposed shared-score quaternion self-attention. (c) QDilated DenseNet module with gated activation.

**RMS Loss.** The root mean square error between the estimated and target waveforms encourages the matching of the energy levels:

$$\mathcal{L}_{\mathrm{RMS}} = \sqrt{\frac{1}{N}\sum_{n=1}^{N}(\hat{x}_n - x_n)^2}. \tag{71}$$

**Complex L1 Loss.** This loss is defined as the L1 error in the real and imaginary parts of the complex spectrogram, which enables simultaneous optimization of the magnitude and phase:

$$\mathcal{L}_{\mathrm{CL1}} = \|\mathrm{Re}[\hat{S}] - \mathrm{Re}[S]\|_1 + \|\mathrm{Im}[\hat{S}] - \mathrm{Im}[S]\|_1. \tag{72}$$

**PESQ Loss (Kim et al., 2019).** For direct optimization of perceptual quality, we incorporated a differentiable approximation of PESQ (Rix et al., 2001) into the loss.

### D.3. Training Configuration

This section describes the preprocessing conditions and settings used for training and evaluation. All the audio recordings were resampled at 16 kHz.

**VoiceBank+DEMAND.** The STFT was computed from the time-domain waveforms using a frame length of $n_{\mathrm{fft}} = 400$, $\mathrm{hop\_length} = 100$, and a Hanning window. During training, segments of 16,000 samples (approximately 1 s) were extracted, and training was performed for 400 epochs with a batch size of 8. We employed the AdamW optimizer (Loshchilov & Hutter, 2019) ($\beta_1 = 0.5$, $\beta_2 = 0.999$) with an initial learning rate of $2 \times 10^{-4}$. To prevent gradient explosion, gradient clipping was applied with a maximum norm of 1.0.

**DNS-Challenge 3 dataset.** We employed a publicly available 16 kHz version of the data. The STFT was computed with a frame length of $n_{\mathrm{fft}} = 512$, $\mathrm{hop\_length} = 256$ and the Hanning window. During training, segments of 32,000

| Hyperparameter | Quaternion | Real-valued |
|---|---|---|
| Encoder blocks | 3 | 4 |
| Growth rate | 64 | 64 |
| Kernel size | $(3, 3)$ | $(3, 3)$ |
| Bottleneck $d_{\text{model}}$ | 64 | 64 |
| Bottleneck $n_{\text{heads}}$ | 4 | 4 |
| Bottleneck $n_{\text{layers}}$ | 2 | 2 |
| Conv kernel size (Conformer) | 31 | 31 |
| Dropout | 0.1 | 0.1 |
| Decoder blocks | 3 | 4 |

*Table 12.* Hyperparameter configuration comparison between the proposed quaternion model and the real-valued baseline.

| Method | Accuracy (%) | Training Time (min) | Speedup |
|---|---|---|---|
| Tay et al. (2019) | $71.09 \pm 0.34$ | $70.56 \pm 1.97$ | – |
| QTN (Yang et al., 2023) | $70.09 \pm 0.17$ | $28.62 \pm 0.22$ | $2.47\times$ |
| Ours | $70.94 \pm 0.24$ | $50.47 \pm 0.88$ | $1.40\times$ |

*Table 13.* Image classification results on CIFAR-100 (Mean $\pm$ Std over 3 runs).

samples (approximately 2 s) were extracted, and training was performed for 150 epochs with a batch size of 16. For validation and testing, performance was evaluated using segments of 160,000 samples (approximately 10 s). We used the dev_testset provided by the DNS-Challenge 3 organizers. We employed the AdamW optimizer ($\beta_1 = 0.5$, $\beta_2 = 0.999$) with a learning rate of $1 \times 10^{-4}$. Similar to VoiceBank+DEMAND, gradient clipping (maximum norm = 1.0) was applied. RTF measurements were conducted on an NVIDIA A100 80GB GPU and an Intel Xeon Gold 6342 CPU with batch size 1 over 448 utterances (6.86k s) under matched implementation conditions for both methods.

**Hyperparameters.** The architectural configurations of QDenseNet, QTransformer, and QConformer are listed in Table 12. All the models used identical STFT preprocessing, loss function definitions, and optimizer settings (AdamW + gradient clipping). The quaternion RMSNorm was applied to queries and keys. Data augmentation was not performed.

## E. Cross-Domain Validation Details

This appendix provides the detailed experimental setup and full results for the cross-domain experiments summarized in Section 5.4, covering image classification on CIFAR-100 (Krizhevsky, 2009) and text classification on SST-2 (Socher et al., 2013).

### E.1. Image Classification (CIFAR-100)

E.1.1. EXPERIMENTAL SETUP

We adopted a configuration in which image features were extracted using a quaternion ResNet, and classification was performed using a quaternion Transformer. The RGB images were encoded as pure imaginary quaternions with the real part set to 0, and each RGB channel was assigned to the imaginary parts (Eq. (73)).

$$Q = 0 + R\mathsf{i} + G\mathsf{j} + B\mathsf{k} \tag{73}$$

**Quaternion ResNet encoder.** We used a 4-stage quaternion ResNet encoder with base channels 64 and 2 residual blocks per stage. The stride pattern was [1,2,2,2], producing a 4×4 grid (16 tokens) before the transformer.

**Transformer bottleneck.** We used 2 layers with $H = 4$ heads, dropout 0.1, embed_dim 64. The total number of parameters in the model was approximately 1.7 M. We used the CIFAR-100 dataset (100 classes, 50,000 training images, and 10,000 test images) and trained it for 100 epochs with a batch size of 128.

| | 0 | 1 | 2 | 3 | | 0 | 1 | 2 | 3 |
|---|---|---|---|---|---|---|---|---|---|
| 0 | 1.000 | -0.248 | -0.124 | -0.063 | 0 | 1.000 | 0.393 | 0.451 | 0.517 |
| 1 | -0.248 | 1.000 | 0.059 | -0.006 | 1 | 0.393 | 1.000 | 0.200 | 0.524 |
| 2 | -0.124 | 0.059 | 1.000 | 0.038 | 2 | 0.451 | 0.200 | 1.000 | 0.401 |
| 3 | -0.063 | -0.006 | 0.038 | 1.000 | 3 | 0.517 | 0.524 | 0.401 | 1.000 |
| | *(a) Random initialization* | | | | | *(b) After training* | | | |

*Table 14.* Gradient norm correlation matrices for the results of Tay et al. (2019) on CIFAR-100. Indices $0, 1, 2, 3$ denote the quaternion components. (a) At random initialization, gradient norms are approximately uncorrelated (mean off-diagonal: $-0.06$). (b) After training on CIFAR-100, moderate correlations emerge (mean off-diagonal: $0.41$). Both experiments use random input data, largely reducing the influence of the input distribution; therefore, the change is attributable mainly to learned weights.

| Method | Accuracy (%) | Latency (ms / batch) |
|---|---|---|
| Tay et al. (2019) | $81.12 \pm 0.70$ | $3.92 \pm 0.12$ |
| Ours | $81.04 \pm 0.30$ | $2.87 \pm 0.07$ |

*Table 15.* Text classification results on SST-2 (mean $\pm$ std over 3 runs). Latency was measured under matched implementation conditions.

We report the mean and standard deviation over three independent runs with different random seeds (42, 123, 456).

We compared the proposed method with two baselines: 1) the quaternion attention of Tay et al. (2019) as a direct baseline, and 2) QTN (Yang et al., 2023) as a reference for convolution-based efficient quaternion architectures. All models were trained under identical conditions. We employed the AdamW optimizer (weight decay = 0.05) (Loshchilov & Hutter, 2019) with a cosine annealing scheduler (with 10 epochs of warmup) to control the learning rate. For data augmentation, we applied Mixup ($\alpha = 0.8$), label smoothing (0.1), and random erasing ($p = 0.25$) techniques.

### E.1.2. Results and Comparison with QTN Results

Table 13 presents the full results, including those of QTN as an additional reference point. QTN achieved the highest speedup ($2.47\times$), which is expected considering its reliance on quaternion convolutions that inherently reduce computational complexity compared to global self-attention (Yang et al., 2023). However, this efficiency comes at the cost of representational power, resulting in a 1.0% accuracy drop compared with the Hamilton-product baseline. Replacing global attention with local convolutions limits the ability of the model to capture the long-range dependencies required for this task.

In contrast, our shared-score method maintains the global context modeling capability of the standard Hamilton-product attention, achieving an accuracy comparable to that of the baseline ($70.94 \pm 0.24\%$ vs. $71.09 \pm 0.34\%$) while delivering a $1.40\times$ training-time speedup. This contrast illustrates that our approach reduces the computational cost without sacrificing the global modeling capabilities essential for high-performance Transformers. (Speedup here refers to the reduction in total training wall-clock time under this setup.)

### E.1.3. Gradient Norm Correlation on CIFAR-100

The gradient norm correlation analysis (Table 14) suggests that the update-magnitude coupling observed in speech enhancement (Table 8) can also arise in image classification. The mean off-diagonal correlation increases from near zero at initialization ($-0.06$) to $0.41$ after training, indicating moderate coupling after learning.

### E.2. Text Classification (SST-2)

### E.2.1. Experimental Setup

To verify that the redundancy claim also holds in the NLP domain, we evaluated the proposed method on the Stanford Sentiment Treebank (SST-2) (Socher et al., 2013), a binary sentiment classification benchmark.

**Quaternion feature representation.** Each input token embedding was reshaped into a quaternion vector by splitting the embedding dimension into four equal parts, with each part assigned to one quaternion component (real and three imaginary parts).

**Model architecture.** We used a quaternion Transformer encoder with 2 layers, $H = 4$ heads, dropout 0.1, and quaternion embedding dimension 64, yielding a total of approximately 100K parameters. A mean-pooled representation was passed to a linear classifier over two classes.

**Training.** All models were trained under matched architecture capacity, batch size, and learning-rate scaling. We used the AdamW optimizer (Loshchilov & Hutter, 2019) with a cosine schedule and a short warmup. We report the mean and standard deviation over 3 independent runs with different random seeds.

### E.2.2. RESULTS

Table 15 reports the SST-2 results. The proposed method preserves accuracy within seed-level variance ($81.04 \pm 0.30\%$ vs. $81.12 \pm 0.70\%$) while reducing inference latency by $26.8\%$ ($1.37\times$ faster). This trend is consistent with the speedups observed on speech enhancement and CIFAR-100, indicating that the efficiency benefit of the shared-score design is not specific to a single modality.

