# OpenReview forum: "Quaternion Self-Attention with Shared Scores"
_ICML.cc/2026/Conference — ICML 2026 regular_

### Official Review · Reviewer_xiWb · 2026-02-25

**Soundness:** 3
**Presentation:** 3
**Significance:** 2
**Originality:** 3
**Overall Recommendation:** 4
**Confidence:** 4

**Summary:**

This paper introduces Shared-Score Quaternion Self-Attention, a simplification of the Hamilton-product-based attention proposed by Tay et al. (2019). Instead of computing four component-wise score matrices and applying separate softmax operations, the method uses a single shared scalar score derived from a quaternion inner product. The authors argue that component-wise independence is structurally redundant and mainly adds computational cost. They validate the approach on single-channel speech enhancement using VoiceBank+DEMAND and DNS-Challenge 3, showing that the shared-score design maintains enhancement quality while delivering 45–61% speedups and around 60% RTF reduction over quaternion attention baselines.

**Compliance With Llm Reviewing Policy:**

Affirmed.

**Final Justification:**

My concerns have been well addressed.

**Key Questions For Authors:**

see Weaknesses.

**Limitations:**

No, see Weaknesses.

**Strengths And Weaknesses:**

Strengths:
1. The efficiency motivation is clearly articulated and backed by a concrete algorithmic simplification.
2. The runtime evaluation goes beyond superficial RTF numbers.
3. The diagnostic analysis is careful and goes beyond raw task metrics.
4. On speech enhancement benchmarks, the proposed method remains competitive with prior quaternion attention while improving RTF by a large margin.

Weaknesses:
1. The evaluation scope feels narrow for the claim being made. The paper positions itself as improving a fundamental building block of quaternion Transformers, yet the main experiments are limited to speech enhancement. Appendix E makes a half-hearted attempt at image classification, but that still doesn't fully address the concern: if the contribution is supposed to be a generally better attention mechanism, you'd want to see it tested on at least one or two other task families, i.e., NLP, time series, or even broader audio tasks. As it stands, the paper doesn't really convince that component-wise independence is unnecessary in general, only that it happens to be unnecessary for this specific task.
2. Comparisons to real-valued baselines are not fully convincing. The paper includes a real-valued Conformer baseline and references several larger-scale models, but it does not clearly establish whether the comparison is fair in terms of compute or capacity. It also leaves unclear what a practitioner should take away: is the message that quaternion attention is preferable to strong modern real-valued architectures, and if so, under what conditions?
3. The conclusion that component-wise softmax is redundant is mostly empirical and may be task- or scale-dependent. The paper shows low inter-component agreement and similar output distributions in the current setup, but these observations do not rule out regimes where component-wise diversity could matter, such as larger models, different training objectives, or longer-context tasks.
4. The RTF and speedup claims are based on a narrow set of hardware and configuration settings, only a high-end GPU (A100), with fixed batch size and sequence length. Measurements on CPU, edge devices, and a kernel-level breakdown are missing. It is also unclear whether the attention implementations are equally optimized for both methods.
5. The claim that "quality does not drop" is supported only on two standard speech enhancement benchmarks. It would be more convincing if the authors included stress tests where attention diversity might actually matter, such as longer sequences, more challenging noise conditions, stronger reverberation, out-of-domain noise, or scenarios with more speakers.

---

> ### Author Rebuttal · Authors · 2026-03-27
>
> Thank you for the detailed and rigorous comments. They were very helpful in improving the paper.
>
> 1. **Scope of the claim**
>
> Our claim is not that component-wise scoring is universally redundant. Rather, we show redundancy under the setting where quaternion linear projections perform pre-mixing, so that component-wise and shared scores lie in the same interaction subspace (Theorem C.1, Corollary C.2). We will clarify that architectures without such quaternion linear layers are outside our scope.
>
> Under this condition, we further validated the method across **speech, vision, and NLP**.
>
> | Domain | KS statistic | Wasserstein distance | Quantile correlation |
> | --- | --- | --- | --- |
> | Speech | 0.0128 | 0.0280 | 0.9953 |
> | Vision (CIFAR-100) | 0.0151 | 0.0203 | 0.9975 |
> | NLP (SST-2) | 0.0407 | 0.0927 | 0.9961 |
>
> These comparisons are computed on the full attention outputs, including softmax and AV multiplication, not only on the attention weights. In all three domains, the quantile correlation exceeds 0.995, suggesting that the extra degrees of freedom in component-wise scoring do not lead to large output differences in practice.
>
> We also confirmed this at the task level on **SST-2** (3 seeds):
>
> | Method | Accuracy (%) | Latency (ms/batch) |
> | --- | --- | --- |
> | Ours | 81.04 ± 0.30 | 2.87 ± 0.07 |
> | Tay et al. (2019) | 81.12 ± 0.70 | 3.92 ± 0.12 |
>
> Our method preserves accuracy within seed-level variance while reducing latency by **26.8% (1.37x)**. This matches the trend observed in speech and vision. We will state more explicitly that larger models, longer contexts, and other objectives remain important future work.
>
> 2. **Comparison with real-valued baselines**
>
> We should have explained this more clearly. The primary comparison target of the paper is Tay et al. (2019), because our contribution is a simplification of quaternion attention, not a claim that quaternion models always outperform modern real-valued architectures. The Real-valued Conformer is included only as a parameter-efficiency reference point. We will revise the main text so this framing is unambiguous.
>
> 3. **Task dependence of the claim**
>
> We agree that component-wise diversity may matter more in larger-scale or longer-context settings. Our evidence therefore supports practical sufficiency in the tested regimes, not universal redundancy. At the same time, the consistent trends across speech, vision, and text suggest that the benefit of the shared-score design is not confined to a single task. We will make this scope explicit in the revision.
>
> 4. **RTF speedup and fairness of runtime comparison**
>
> Following your comment, we revisited implementation fairness and found an asymmetry in the original runtime comparison: the Tay et al. baseline used a sequential loop over split heads, whereas our method used a batched implementation. We therefore re-ran both methods under the same implementation policy.
>
> | Environment | Model | Tay et al. RTF | Ours RTF | RTF reduction | Speedup |
> | --- | --- | --- | --- | --- | --- |
> | GPU (A100) | Transformer | 0.0192 | 0.0107 | 44.3%↓ | 1.79x |
> | GPU (A100) | Conformer | 0.0202 | 0.0157 | 22.3%↓ | 1.29x |
> | CPU (Xeon Gold 6342) | Transformer | 0.594 | 0.249 | 58.1%↓ | 2.39x |
> | CPU (Xeon Gold 6342) | Conformer | 0.610 | 0.259 | 57.5%↓ | 2.36x |
>
> The GPU gains are smaller than originally reported, but the efficiency improvement remains consistent, and the CPU results show that it is not specific to a single A100 setting.
>
> This correction affects all wall-clock comparisons measured under the asymmetric implementation. In the revision, we will update Tables 3, 8, and 12 accordingly. The corrected single-layer speedups in Table 8 are 1.27x / 1.65x / 1.86x / 2.08x for sequence lengths 512 / 1024 / 2048 / 4096, respectively; MACs remain unchanged because they are theoretical values. For Table 12, the corrected CIFAR-100 training time of Tay et al. is 70.56 min, yielding a 1.40x speedup. We appreciate your comment because it helped us identify and fix this fairness issue.
>
> 5. **Quality evaluation**
>
> We agree that broader robustness evaluation is important. DNS-Challenge 3 is already a large-scale benchmark, including 760 h of clean speech, 181 h of noise, and about 118,000 room impulse responses, covering diverse SNR, reverberation, and multilingual conditions. In this setting, our method maintains quality comparable to or better than the baseline, which provides some evidence of robustness. However, we have not yet evaluated longer sequences, extreme reverberation, OOD noise, multi-speaker settings, edge devices, or kernel-level runtime breakdowns. We will state these clearly as limitations in the revised manuscript.
>
> Finally, we thank you again for the constructive comments, especially on runtime fairness, which led us to present a more accurate and fair comparison.

---

> > ### Author Rebuttal · Reviewer_xiWb · 2026-04-02
> >
> > Thank you for the rebuttal. I have two remaining clarification questions. First, since the rebuttal now makes the claim more conditional, namely that the redundancy argument is intended for the setting where quaternion linear projections already induce substantial pre-mixing, will the revised manuscript explicitly reflect this narrower scope in the abstract, introduction, and conclusion? At present, the paper still reads more broadly than the rebuttal now supports. Second, since the rebuttal identifies an implementation asymmetry in the original runtime comparison, will the revision systematically update all headline efficiency claims, tables, and discussion so that the reported speedups consistently reflect fair wall-clock comparisons rather than the original asymmetric implementation? More broadly, I also encourage the authors to make clearer in the revision which regimes are actually supported by current evidence, and which settings such as larger models or longer-context tasks remain outside the validated scope.

---

> > > ### Author Response · Authors · 2026-04-02
> > >
> > > Thank you for the follow-up. We answer both questions directly.
> > >
> > > **Q1: Will the revised manuscript explicitly reflect the narrower scope in the abstract, introduction, and conclusion?**
> > >
> > > Yes. We will make this narrower scope explicit throughout the manuscript. Concretely, we will state that our redundancy claim is intended for architectures where queries and keys are generated by quaternion linear projections that already induce component pre-mixing (Theorem C.1), rather than as a universal statement about all quaternion attention designs. Although this is a narrower claim, it still covers the standard Quaternion Transformer setting considered in prior work, where queries and keys are produced by quaternion linear projections.
> > >
> > >
> > > Specifically:
> > > (i) the abstract will qualify the “same interaction subspace” statement with this condition;
> > > (ii) Contribution 1 in the introduction will be revised to read “under the pre-mixing condition induced by quaternion linear projections (Theorem C.1)”; and
> > > (iii) the conclusion will explicitly state that larger models, longer-context tasks, and architectures without such pre-mixing layers remain outside the validated scope of this paper and are important future work.
> > >
> > > We will also revise the broader design-language in Section 5.3 so that the manuscript consistently frames our claim as conditional on architectures where quaternion linear projections already induce component pre-mixing, rather than as a universal design rule for hypercomplex attention.
> > >
> > > **Q2: Will the revision systematically update all headline efficiency claims, tables, and discussion to reflect fair wall-clock comparisons?**
> > >
> > > Yes. We will update all wall-clock claims consistently so that they reflect the fair re-measurement under matched implementation conditions.
> > >
> > > | Location | Old (asymmetric) | Corrected (fair) |
> > > |---|---|---|
> > > | Abstract | “45–61%” | “up to 44.3% GPU / 58.1% CPU RTF” |
> > > | Conclusion | “45–61% faster inference” | “up to 44.3% GPU / 58.1% CPU RTF” |
> > > | Section 5 opening | “approximately 60%” | “GPU: 44.3%↓ (QTransformer), 22.3%↓ (QConformer); CPU: 58.1%↓, 57.5%↓” |
> > > | Table 3 — QTransformer GPU | 60.9%↓, 2.6× | 44.3%↓, 1.79× |
> > > | Table 3 — QConformer GPU | 44.9%↓, 1.8× | 22.3%↓, 1.29× |
> > > | Table 3 (new rows) — CPU | — | QTransformer 58.1%↓, 2.39× / QConformer 57.5%↓, 2.36× |
> > > | Table 8 — T=512/1024/2048/4096 | 4.29× / 3.53× / 1.92× / 2.08× | 1.27× / 1.65× / 1.86× / 2.08× |
> > > | Table 12 — CIFAR-100 | Tay: 99.98 min → Speedup: 1.98× | Tay: 70.56 min → Speedup: 1.40× |
> > >
> > > MACs in Table 8 are theoretical and remain unchanged.
> > >
> > > In addition, we will revise all corresponding narrative text in Section 4.5, the opening of Section 5, the conclusion, and Appendices C.4/E.2 so that every wall-clock statement consistently reflects the corrected fair comparison.
> > >
> > > As a limitation, the current evidence supports practical sufficiency only in the tested regimes; we have not yet validated larger models, longer-context tasks, architectures without quaternion pre-mixing projections, or more challenging robustness settings such as extreme reverberation, out-of-domain noise, and multi-speaker scenarios, and we will state these explicitly in the revised manuscript.
> > >
> > > Finally, to align the manuscript with the scope actually supported by the current evidence, we will promote the SST-2 task results and the three-domain output-distribution comparison (speech / vision / NLP) from the rebuttal into the main paper.

---

### Official Review · Reviewer_iuTH · 2026-03-07

**Soundness:** 2
**Presentation:** 3
**Significance:** 2
**Originality:** 3
**Overall Recommendation:** 4
**Confidence:** 4

**Summary:**

The paper proposes a shared-score quaternion self-attention mechanism in quaternion neural networks, which has been shown to be efficient for modeling multidimensional speech signals. The main idea is to compute a attention score using the scalar product of quaternion inner product and share this score across all quaternion components instead of computing independent scores for each component. This shared score mechanism will reduce the redundant computation and enforces consistent attention distributions across quaternion channels.

**Compliance With Llm Reviewing Policy:**

Affirmed.

**Key Questions For Authors:**

1. Explain the inherent advantages of inner product, which ignores the inter-latent interactions across multiple quaternion components.

2. To ensure the experiments reproducibility, the details for data preparation and initial settings must be clearly described. In particular, the complete name of  STFT transformation must be specified in the first time.

3. Please clearly highlight in Tables 2 and 3 the computational times advantages of the proposed method over the QTransformer (Tay et al., 2019) method.

**Limitations:**

yes

**Strengths And Weaknesses:**

Strengths
1) Problem Importance: The paper proposes a shared-score quaternion self-attention mechanism in quaternion neural networks, which has been shown to be efficient for modeling multidimensional speech signals. The analysis shows that the component-wise and shared scores lie in the same interaction subspace, which would be beneficial for quaternion neural networks. The problem addressed in the paper is meaningful.

2) Novelty: The main idea is to compute a attention score using the scalar product of quaternion inner product and share this score across all quaternion components instead of computing independent scores for each component. This shared score mechanism reduces redundant computation and enforces consistent attention distributions across quaternion channels. While the idea is conceptually simple, it provides a clear modification to existing quaternion attention mechanisms and leads to a computationally lighter formulation.

3) Computational efficiency improvement: The method reduces score computation multiplications by approximately 75 percent and reduces softmax operations from four to one. The reported inference speedup is substantial.

4) Theoretical insight:The paper provides an analysis showing that the component wise and shared attention scores lie in the same bilinear interaction subspace. This offers an interesting perspective on the redundancy of independent component attention.

Weaknesses
1) Limited explanation for the benefit of the scalar product: The proposed attention score is essentially the scalar product, which is closely related to standard real-valued dot product attention. While it utilizes the Hamilton product, which better encodes the inter-latent interactions across multiple quaternion components in the quaternion setting.

2) Theoretical claim not fully justified: Theorem C.1 and Corollary C.2 show that the four score components lie in the same bilinear interaction subspace. However, the attention mechanism applies a nonlinear softmax to each score. Therefore, identical linear interaction spaces do not necessarily imply equivalent expressive power of the resulting attention distributions. Independent component-wise softmax operations could still produce distinct attention patterns that cannot be reproduced by a single shared score.

3) Limited experiments: Experiments are conducted on only two speech datasets, which is insufficient to demonstrate the diversity of the dataset. There are no explicit results to show the benefit regarding computational cost and time in Tables 2 and 3.

---

> ### Author Rebuttal · Authors · 2026-03-27
>
> Thank you for the helpful comments. The points you raised are all important, and in the revised version we will clarify the scope of the theory, the experimental coverage, reproducibility, and the presentation of runtime results. We respond point by point below.
>
>
> 1. **On the design advantage of the shared scalar score and the scope of Theorem C.1**
>
>    The shared scalar score in our method,
>
>    $
>    S_{ij}=\mathrm{Re}\left(\mathbf{Q}_i \otimes \mathbf{K}_j^{\dagger}\right)
>    $
>
>    does not treat the cross-component terms of the Hamilton product separately in the score computation. However, because the quaternion linear projection already integrates multiple components into each projected component, as shown by Theorem C.1 / Corollary C.2, Tay et al.'s component-wise score and our shared score belong to the same bilinear interaction subspace. Therefore, our method does not discard cross-component interactions; rather, it is a design that separates the representation of interactions into the projection side and the generation of the attention map into the scoring side.
>
>    At the same time, as you correctly point out, Theorem C.1 is a pre-softmax result and does not guarantee complete functional equivalence after the softmax. For that reason, in addition to theory, the paper provides gradient-correlation analysis and final attention-output comparisons (Figure 3 / Figure 4) as complementary evidence. In Figure 3, we report KS statistic = 0.0128, Wasserstein distance = 0.028, and quantile correlation = 0.995; in Figure 4(b,c), we observe output correlations of $r = 0.78$--$0.88$ between the two methods, where $(Q, K, V)$ are extracted from two independently trained models—one trained with our method and one with Tay et al.—and each is applied to its own attention computation. The high correlation between the resulting outputs, despite independent training, further supports the view that the two formulations converge to functionally similar representations.
>
> 2. **Response to the concern about reproducibility**
>
>    Thank you for this suggestion. We note that many of these details are already provided in Appendix D.3; however, we recognize that the lack of explicit cross-references from the main text makes them difficult to locate. In the revised version, we will add clear pointers from the main text to the relevant appendix sections, and additionally consolidate the following details to further improve readability: (i) explicitly define STFT as Short-Time Fourier Transform at first mention, and confirm the FFT size, window length, and hop length are clearly stated in the main text; (ii) describe the data preprocessing and the train/validation/test split; (iii) specify the optimizer, learning-rate schedule, warmup, batch size, number of epochs, and number of seeds; and (iv) detail the runtime/RTF measurement conditions (hardware, batch size, sequence length, and measurement protocol). We believe these additions will make the paper substantially easier to follow from a reproducibility perspective.
>
> 3. **On runtime**
>
>    Our presentation of runtime was also not sufficiently clear. Based on re-measurement under fair implementation conditions, we will add the RTF results on VoiceBank+DEMAND to Table 2 and update the runtime comparison in Table 3 so that it is clearer. In the revised version, we will state more explicitly in both the table and the main text that we observed runtime improvements over QTransformer while maintaining comparable accuracy.

---

> > ### Author Rebuttal · Reviewer_iuTH · 2026-04-02
> >
> > I would like to thank the authors for their response. I am keeping my current score.

---

### Official Review · Reviewer_fzNk · 2026-03-13

**Soundness:** 3
**Presentation:** 3
**Significance:** 3
**Originality:** 4
**Overall Recommendation:** 4
**Confidence:** 3

**Summary:**

This paper proposes a shared score quaternion self attention method. Instead of computing four separate component-wise attention scores and softmax operations, it uses a single real valued score from the quaternion inner product and shares one attention map across all quaternion components. This simpler design keeps the important interactions between tokens while reducing computation. The method is tested on speech enhancement tasks, resulting in much faster inference.

**Compliance With Llm Reviewing Policy:**

Affirmed.

**Final Justification:**

The rebuttal has fully addressed my concerns. I will keep my recommendation score is 4.

**Key Questions For Authors:**

How broad is the claim of redundancy beyond speech enhancement?

**Limitations:**

Yes.

**Strengths And Weaknesses:**

Strengths:
1. Soundness: this is a solid paper with a clear idea. The authors show that quaternion attention does not need four separate attention maps, and instead propose using one shared real-valued score from the quaternion inner product. That idea is simple, and practically useful because it cuts score multiplications and reduces the number of softmax operations from four to one.

2. Presentation: the paper is mostly easy to read and the method is well motivated

3. Significance: I think the problem is meaningful. Quaternion models are often sold as efficient and structural, so it is useful to ask whether their attention mechanism is doing extra work that is not actually needed. The paper gives a practical answer: a simpler shared-score design can keep most of the benefit while making inference much faster.

4. Originality: the idea is useful and novel in my opinion.



Weaknesses:
1. Soundness: the authors claims that component-wise independence is structurally redundant, and the evidence is suggestive, but most of the validation is still centered on speech enhancement.

In addition, the paper’s main theory shows that the shared score and the four component-wise scores live in the same interaction subspace, but that is weaker than showing they are equally expressive after the softmax and the rest of the network. In other words, “same bilinear subspace” does not automatically mean “same functional capacity” once the nonlinear attention pipeline is applied.

2. Significance: the limitation is that the demonstrated impact is still fairly narrow. Most of the strong evidence is in speech enhancement, and the broader claim about hypercomplex attention design is promising but not yet fully established across very different tasks.

---

> ### Author Rebuttal · Authors · 2026-03-27
>
> We appreciate your positive assessment of the simplicity and practicality of our method.
>
> We understand your main concerns to be twofold: (1) how far the claim of "redundancy" can be extended beyond speech enhancement, and (2) that Theorem C.1 only characterizes the pre-softmax structure and does not guarantee full functional equivalence after the softmax.
>
> 1. **How far does the redundancy claim extend beyond speech enhancement?**
>
>    We conducted an additional experiment on **NLP (SST-2)**, as in Tay et al., and also performed an **output-comparison analysis across three domains: speech, vision, and NLP**.
>
>    **Task performance and inference efficiency (SST-2, 3 seeds):**
>
>    | Method | Accuracy (%) | Latency (ms/batch) |
>    | --- | --- | --- |
>    | Ours | 81.04 ± 0.30 | 2.87 ± 0.07 |
>    | Tay et al. (2019) | 81.12 ± 0.70 | 3.92 ± 0.12 |
>
>    We maintained comparable accuracy within seed-level variance while reducing inference latency by 26.8% (1.37× speedup).
>
>    Next, we extended the output-distribution comparison from Figure 3 to all three domains.
>
>    **Similarity of output distributions between trained models (extension of Figure 3):**
>
>    | Domain | KS statistic | Wasserstein distance | Quantile correlation |
>    | --- | --- | --- | --- |
>    | Speech enhancement | 0.0128 | 0.0280 | 0.9953 |
>    | Vision (CIFAR-100) | 0.0151 | 0.0203 | 0.9975 |
>    | NLP (SST-2) | 0.0407 | 0.0927 | 0.9961 |
>
>    **In all three domains, the quantile correlation exceeds 0.995**, confirming that the shapes of the output distributions under the two formulations are extremely similar. The KS statistic and Wasserstein distance are somewhat larger for NLP, but their absolute values remain small, and task accuracy is still comparable within seed-level variance. Taken together, these results indicate across multiple domains that the additional degrees of freedom of component-wise scoring do **not translate into meaningful practical performance differences**. We will promote these results to the main paper in the revised version.
>
> 2. **Does belonging to the same bilinear subspace not necessarily imply the same post-softmax functional capacity?**
>
>    You are correct. Theorem C.1 / Corollary C.2 concern the pre-softmax score structure and do not by themselves guarantee complete functional equivalence after the softmax. Indeed, Tay et al. maintains four independent post-softmax attention maps, one for each quaternion component, whereas our method uses a single shared map; the post-softmax structures of the two methods are therefore intrinsically asymmetric. For this reason, establishing strict theoretical equivalence via a one-to-one correspondence at the post-softmax level is not the natural target in our comparison setting.
>
>    Instead, in this work we chose to evaluate the final post-softmax behavior empirically. Specifically, Figure 3 compares the final output distributions of the two methods after the full attention computation, including the softmax and $AV$ multiplication. Figure 4(b,c) reports output correlations between the two formulations, where $(Q, K, V)$ are extracted from **separately trained models** and each applied to their respective algorithm. The fact that the outputs remain highly correlated despite independent training further supports the view that the additional degrees of freedom in the four-map formulation do not lead to meaningfully different representations. These results suggest that, at least in the settings examined in this study, the extra degrees of freedom associated with four attention maps do not manifest as large differences in the final output. Accordingly, the claim of this paper is not that the two formulations are "theoretically fully equivalent," but rather that, taken together, the structural overlap at pre-softmax (Theorem C.1) and the empirical results at the post-softmax and task levels indicate that the shared-score design is practically sufficient and serves as a more efficient alternative.

---

> > ### Author Rebuttal · Reviewer_fzNk · 2026-04-03
> >
> > Thank you for your response. The rebuttal addressed my concerns. I am keeping my current score of 4.

---

### Official Review · Reviewer_Fg1x · 2026-03-19

**Soundness:** 2
**Presentation:** 2
**Significance:** 2
**Originality:** 2
**Overall Recommendation:** 4
**Confidence:** 3

**Summary:**

The paper focuses on Quaternion neural networks, in particular it attempts to make self-attention more efficient for QNNs. In particular, instead of computing component-wise scores, it computes a single real-valued shared score. This reduces the score-computation by almost one-fourth. The method is applied to speech enhancement where it is shown to provide benefits over previous Quaternion transformers in terms of both performance and compute. It also provides justification of wny shared score is sufficient.

**Compliance With Llm Reviewing Policy:**

Affirmed.

**Final Justification:**

Some of my concerns were addressed.  Keeping positive acceptance score as is.

**Key Questions For Authors:**

Please address the weaknesses.

**Strengths And Weaknesses:**

Quaternion neural networks are a bit outside of my core expertise, and I am not super familiar with the literature and latest developments.

Strengths
-- The method collapses attention computation over each component to just a single shared score. I believe applications where QNNs are useful, it might make sense to have shared attention maps for tighter coupling
-- The formulation does lead to good reduction in compute. Real multiplications and softmax operations are reduced by a factor of 4.
-- There is a good attempt (Section 5)  to provide justification for the proposed method.

Weaknesses
-- I am not sure what drove the application to speech enhancement. The paper does not attempt to provide any justification on why SE is a good application. The paper is treating (Tay et al., 2019) as the base, and I am wondering why not stick with the same set of applications as (Tay et al., 2019). At least some experiments following  (Tay et al., 2019) would have made sense.
-- It would be better to provide some context around speech enhancements and QNNs, why it makese sense to explore it even though QNNs are not needed for state-of-the-art performance or low-compute.
-- (Tay et al., 2019)  seems like a really old work to built upon. Are there no recent developments on QNNs.

---

> ### Author Rebuttal · Authors · 2026-03-27
>
> Thank you for the helpful comments. We respond to your concerns in turn.
>
> - **Why did we use speech enhancement as the primary evaluation task, rather than the original tasks in Tay et al.?**
>
>   The primary reason we chose speech enhancement as our main testbed is that **quaternion feature construction naturally aligns with complex STFT representations**. In speech enhancement, correlated components such as the real part, imaginary part, and magnitude must be modeled jointly, and phase is also crucial for perceptual quality. For this reason, we considered speech enhancement to be a natural primary testbed for examining whether component-wise attention is truly necessary in quaternion attention.
>
>   In addition, latency and RTF are important in real-world speech enhancement systems, which allows the value of attention efficiency, as targeted in this work, to be evaluated directly. In this sense as well, speech enhancement is well aligned with the goal of our study.
>
>   At the same time, to address your concern, we also conducted additional experiments on **text sentiment classification (SST-2)** following Tay et al. Under matched architecture capacity (approximately 100K parameters), batch size, and learning-rate scaling, we obtained the following results averaged over 3 seeds:
>
>   | Method | Acc. (%) | Latency (ms) |
>   | --- | ---: | ---: |
>   | Ours | $81.04 \pm 0.30$ | $2.87 \pm 0.07$ |
>   | Tay et al. (2019) | $81.12 \pm 0.70$ | $3.92 \pm 0.12$ |
>
>   That is, our method achieves comparable accuracy within seed variance while improving inference latency by $1.37\times$ ($26.8\%$ reduction). This trend is also consistent with what we observed in speech enhancement and CIFAR-100, suggesting that the effect of the proposed method is not limited to speech enhancement alone. We will reflect this in the revised version.
>
> - **Is Tay et al. (2019) too old? How does this work relate to more recent QNN research?**
>
>   Thank you for raising this point. It is not the case that there has been no recent work on QNNs / hypercomplex networks. In fact, we are aware of more recent studies such as Chen et al. (2022) [1], Yang et al. (2023) [2], Zhou et al. (2024) [3], and Mukhopadhyay et al. (2024) [4]. That said, we chose Tay et al. (2019) as our primary comparison target because it is the formulation that **directly instantiates the Hamilton-product-based quaternion self-attention scoring that our work aims to simplify**. Many subsequent studies instead focus on quaternion convolution, rotational embeddings, or parameter sharing / pruning using quaternion algebra, rather than directly comparing designs in which the **attention scores themselves are computed via Hamilton products**. Therefore, we believe that Tay et al. (2019) is the most direct prior formulation for our problem setting.
>
> ---
>
> **References**
>
> [1] Chen, W., Wang, W., Peng, B., Wen, Q., Zhou, T., and Sun, L. Learning to rotate: Quaternion transformer for complicated periodical time series forecasting. *KDD '22*, pp. 146–156, 2022. doi: 10.1145/3534678.3539234.
>
> [2] Yang, X., Cao, W., Lu, Y., and Zhou, Y. Qtn: Quaternion transformer network for hyperspectral image classification. *IEEE Transactions on Circuits and Systems for Video Technology*, 33(12):7370–7384, 2023. doi: 10.1109/TCSVT.2023.3283289.
>
> [3] Zhou, Z., Huo, Y., Huang, G., Zeng, A., Chen, X., Huang, L., and Li, Z. Qean: Quaternion-enhanced attention network for visual dance generation. *Vis. Comput.*, 41(2):961–973, 2024. doi: 10.1007/s00371-024-03376-5.
>
> [4] Mukhopadhyay, A., Joshi, R. B., Tiwari, N., and Mishra, S. Transformers at a fraction. In Northern Lights Deep Learning Conference 2025, 2024.

---

> > ### Author Rebuttal · Reviewer_Fg1x · 2026-04-03
> >
> > I found the rebuttal satisfactory. Having said that, I still believe more justification for SE is needed. Also, some more relevant methods in SE should be compared with. Moreover, if there are more recent QNN methods then they should be brought into picture instead of completely keeping them aside.

---

> > > ### Author Response · Authors · 2026-04-06
> > >
> > > Thank you for the acknowledgement and for the additional suggestions. In the revision, we will further strengthen the paper by clarifying the motivation for applying the method to speech enhancement, incorporating more relevant recent SE comparisons, and positioning recent QNN works more clearly in the related work discussion.

---

### Decision · Program_Chairs · 2026-04-30

**Decision:**

Accept (regular)

**Comment:**

The paper considers Quaternion Neural Networks. Existing formulations compute 4 distinct scores, and this paper proposes using a single shared score instead. This reduces the computational cost by about one-fourth. The reviewers found the contribution and improvement over the prior work of Tay et al. (2019) to be solid. However, there wasn't too much enthusiasm either from any of the reviewers and  concerns were raised that the results are a bit narrow, and it would be good to have more detailed evaluations for speech enhancement and more broadly to other tasks. The author response which discussed some more empirical findings helped address some of these concerns.

I generally agree with the reviewers assessments, while the paper appears to be a good improvement over Tay et al. (2019), more detailed evaluations will help the work be more impactful. Overall, I recommend the paper be accepted assuming there is space in the program.